# A microbotanical and microwear perspective to plant processing activities and foodways at Neolithic Çatalhöyük

**Carlos G. Santiago-Marrero**[1]*, **Christina Tsoraki**[2], **Carla Lancelotti**[1,3],
**Marco Madella**[1,3,4]

1 CaSEs—Department of Humanities, Universitat Pompeu Fabra, Barcelona, Spain, 2 School of Archaeology and Ancient History, University of Leicester, Leicester, United Kingdom, 3 Institució Catalana de Recerca i Estudis Avançats (ICREA), Barcelona, Spain, 4 School of Geography, Archaeology and Environmental Studies, The University of the Witwatersrand, Johannesburg, South Africa

* carlos.santiago@upf.edu

## Abstract

Çatalhöyük is a renowned archaeological site in central Anatolia, best known for its Neolithic occupation dated from 7100 to 6000 cal BC. The site received worldwide attention early on for its large size, well-preserved mudbrick architecture, and elaborate wall paintings. Excavations at the site over almost three decades have unearthed rich archaeobotanical remains and a diverse ground stone assemblage produced by what once was a vibrant farming community. The study presented here adds to our understanding of crops and plant processing at Çatalhöyük by integrating phytoliths and starch analyses on grinding implements found at three domestic contexts attributed to the Middle (6700–6500 cal BC) and Late (6500–6300 cal BC) period of occupation. Our results reveal a rich microbotanical assemblage that testifies the use of a wide range of geophytes and wild seasonal resources previously unknown at the site. Moreover, by comparing results from the microbotanical proxies and microscopic wear patterns on artefacts, we are also able to discern various plant processing practices the analysed artefacts were employed for. In sum, this work further expands our understanding of plants and crop processing activities performed by the inhabitants of Neolithic Çatalhöyük.

## 1.0 Introduction

Archaeobotanical remains are the best proxies for studying the emergence of prehistoric agricultural practices. The informative potential of such remains, coupled with ethnographic and ecological models, allowed researchers to identify diverse plant processing strategies of past societies [1–5]. Within these strategies, how plants are transformed into food and crafts have always been central themes. Traditionally these issues have been approached by recovering carbonized macrobotanical remains such as seeds and other inflorescence anatomical parts (chaff, spikelets, forks, etc.) [6, 7]. However, it is only by connecting artefactual assemblages and plant remains that we can gain a broader and more complete picture of plant-processing activities [8, 9]. Grinding implements are among the oldest and most numerous artefacts

**Funding:** This work was funded by the following: The Çatalhöyük Research Project (http://www. catalhoyuk.com/); CaSEs – Quality Research Group of the Catalonian Government SGR-212 (https:// www.upf.edu/web/cases); Raindrops ERC starting grant (Grant agreement ID: 759800) CL, MM; Ph. D. Scholarship (PIPF-UPF-PhD), Department of Humanities Universitat Pompeu Fabra (https:// www.upf.edu/web/humanitats/) CG.S-M; CRAFTS Marie-Curie Intra-European Research Fellowship (Grant agreement no. PIEF-GA-2012-328862) CT.

**Competing interests:** The authors have declared that no competing interests exist.

found in archaeological sites that can be directly associated with plant processing and food production activities [10–13]. Indeed, it has been suggested that an increase in grinding tools and the establishment of cereal-based economies in Southwest Asia are directly correlated [11, 14–16].

Except for some exceptional finds (e.g., [12, 17]), it is generally difficult to make a direct link between macrobotanical remains and grinding implements. Where possible, such a link relies mainly on contextual associations of macrobotanical remains and archaeological finds and on an understanding of taphonomic conditions of depositional practices (primary vs. secondary deposition, but also intentional staged depositions, see [18]) to interpret plant processing areas and related practices within settlements [16, 19–21]. However, the often mobile nature of many grinding tools and the wide spectrum of activities they may have been used for in the past that included processing cereals, underground storage organs (USOs), leaves and shoots, and non-food related products such as pigments and hides, further complicates such contextual associations [22–25]. In addition, vegetal remains such as USOs, leaves and shoots are often absent in the macrobotanical record due to poor preservation via carbonization or because they could have been eaten or processed raw and as such are less prone to accidental charring [13, 16, 24, 26–30].

Significant advances in the functional analysis of grinding tools during the last two decades have revealed a wealth of information about the function of tools and subsistence practices more broadly [22, 25, 31–35]. Moreover, grinding tools from key archaeological sites dated to the Paleolithic and Neolithic period have been sampled for microbotanical remains and more recently there has been an attempt to integrate the results of microwear studies with the study of inorganic and organic micro-residues such as phytoliths and starch (i.e. [24, 36–46]).

Phytoliths originate from the deposition of opal silica in plant cells and cell walls creating casts of the cells or entire tissues. Grasses are the most prolific producers but phytoliths are found in many other plant groups with species of economic interest [47]. Since phytoliths are inorganic, they do not rely on charring for their preservation and are directly integrated into the sediment upon plant decay. Once in the sediment, they remain relatively stable, providing a reliable proxy for investigating aspects of ecology and human-plant interactions [48–52]. One of the most useful aspects of phytoliths is the possibility to identify their plant sources to the family, genus and sometime species level, as well as differentiating the input of different anatomical parts such as culms, leaves, and inflorescences, that will otherwise be nearly invisible in the archaeobotanical record [10, 47, 48, 53–55]. Starch is a carbohydrate made of polymers amylose and amylopectin. It is particularly abundant in seeds and underground storage organs, acting as energy storage for plants [56, 57]. Starch grains proved to be a valuable tool for identifying plant taxa that are rarely preserved through other means or for which processing creates a bias for their survival as macrobotanical remains (i.e., [58–63]). Despite being susceptible to biological and chemical degradation, starch grains (like phytoliths) can survive in the archaeological record for thousands of years under stable conditions, such as in the depressions and crevices present on the use-faces of grinding and pounding tools, in ceramic vessels, in human dental calculus, and even in sediments and can act as a record of an artefact's activity history [43, 57, 64–66]. This preservation pathway enables the study of plant remains originating directly from the implements used for their processing and allows the identification of plants and plant parts that would otherwise remain invisible in the archaeobotanical record. The research presented here integrates for the first time phytoliths, starch analyses on grinding implements at Çatalhöyük, one of the key Neolithic sites in Southwest Asia. The results are then considered in relation to the microwear evidence, therefore presenting a more integrated understanding of tool utilization and use history. By analyzing microbotanical remains trapped on the surfaces of upper and lower grinding tools, we discuss aspects of plant use and

related practices in household contexts attributed to the Middle (6700–6500 cal BC) and Late Period (6500–6300 cal BC) of occupation.

## 1.1 Çatalhöyük, an early farming community in the Konya plain

Located in the Konya plain in central Anatolia, Çatalhöyük consists of two mounds: the East Mound which was continuously occupied from 7100BC to 6000 cal BC [67, 68], and at short distance the West Mound, where the Chalcolithic settlement was established around the beginning of the sixth millennium and was occupied until 5500 cal BC [69] (Fig 1). First excavated by J. Mellaart in the 1960s, in 1993 Çatalhöyük became the focus of a program of excavations directed by I. Hodder that lasted 25 years [70, 71]. During its habitation, the site was located in a highly dynamic and seasonal landscape dominated by the Çarşamba river, which originates in the Taurus Mountains and flows to the north towards the Konya Plain. At Çatalhöyük the river runs through the East (Neolithic) and West (Chalcolithic) mounds forming multiple channels, branches and a combined set of seasonal wetlands, ponds and islands [72–74]. This alluvial landscape provided environmental niches dominated by marshy plants such as reeds (*Phragmites* spp.) and sedges (*Cyperus* spp.), combined with suitable dry areas for farming [72–79].

With a vast record covering almost three decades of continuous research, Çatalhöyük is a Neolithic site with the most extensive and best-studied archaeobotanical assemblage in the central Anatolian region [76, 80, 81]. This assemblage is dominated by cereals such as glume wheat (*Triticum* spp.) and barley (*Hordeum* sp.), followed by pulses such as lentils (*Lens* sp.), peas (*Pisum* sp.) and chickpeas (*Cicer* sp.). Wild resources are also present in the form of fruits and nuts such as acorns (*Quercus* sp.), pistachios (*Pistacia* sp.) and almonds/plums (*Prunus* spp.). There is also evidence for the use of underground storage organs (USO) from the sedge club-rush (*Bolboschoenus glaucus*) and the collection and storage of wild seeds (i.e. [82, 83]), including wild mustard (*Descurainia* spp.—see [76, 80] for a detailed description). This assemblage suggests a mixed and flexible agricultural economy based on the seasonal cultivation of cereals and pulses, combined with the collection of available wild resources such as fruits and nuts [21, 76, 80, 84, 85].

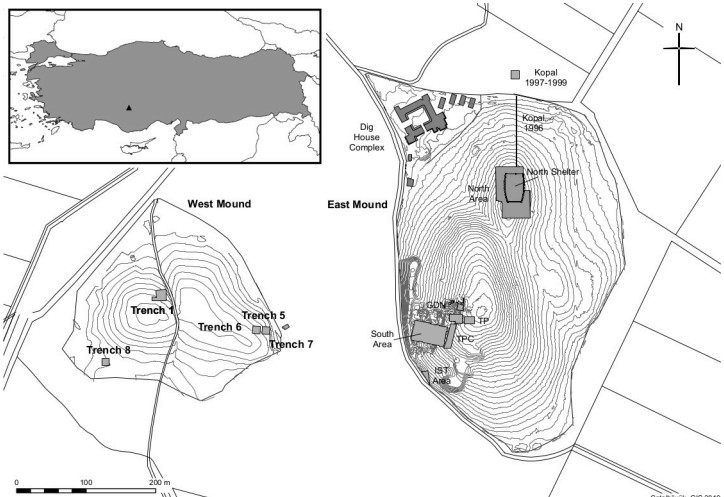

**Fig 1. Plan of Çatalhöyük.** Site's location and plan map of the west mound and the north and south areas on the east mound (after Camilla Mazzucato; Çatalhöyük Research Project).

The archaeobotanical assemblage also shows evidence for on-site post-harvesting processing mostly related to cereal cleaning and storage, as well as some forms of final stage processing of glume wheat, which was probably stored as spikelets to be processed when needed, while barley and free-threshing wheat were likely cleaned off-site and stored as clean grains [76, 80, 81, 85]. Further evidence suggests that small-scale winnowing and sieving could have occurred at a household level [86], and preserved fragments of pea pods seems to indicate that pulses were also processed on-site [21, 80, 81]. Further insights about plant food consumption and meals are provided by the analysis of amorphous charred fragments, some containing cereal grains, interpreted as bread-like products and porridge [84], and others with cereals, legumes and fish [80, 87]. Apart from the macrobotanical evidence, questions about food production, and the location and scale of plant processing activities at Çatalhöyük, have been approached through the technological, contextual and microwear analyses of the site's grinding implements [18, 20, 23, 88–90]. Although phytolith and starch grain analyses had already been carried out at the site, these focused on the investigation of plant raw materials and disposal practices [79, 91–93]. Therefore, apart from some exploratory studies [93, 94], the grinding tools at Çatalhöyük have never been subjected to an integrated microbotanical and microwear study until now.

## 2.0 Materials and methods

### 2.1 The buildings and the grinding tools

Excavations at Çatalhöyük brought to light a wide range of stone implements that includes different types of grinding and abrading tools (e.g., grinders, querns, abraders, pestles, palettes and polishers), stone axes and adzes, hammerstones, centrally perforated spheres ('maceheads'), abraded pigment nodules along with roughouts, preforms and debitage associated with the production of ground stone artefacts [23, 88, 90]. Detailed technological, typo-morphological and contextual study of the Çatalhöyük ground stone assemblage, the results of which are presented in detail elsewhere [88, 90], provides important insight in the life histories of these artefacts and the social practices that surround their use and discard by this Neolithic community. Grinding tools formed part of a suite of stone implements that constituted the regular household toolkit at Çatalhöyük. Grinding tools of different sizes and modes of operation were in use at the site throughout the occupation of the settlement, while during the Middle and Late periods fixed grinding installations are found across different buildings [90]. Querns are on average 342.06 mm long and 223.41 mm wide, and their weight ranges from 2 to 54.6 kg. The grinding tool assemblage is characterized by a high level of fragmentation [18, 88], which over-exaggerates the actual number of grinding tools in use at Neolithic Çatalhöyük. The number of complete specimens unearthed during the Çatalhöyük Research Project excavations amounts to 27 querns and 53 grinders. In morphological terms, querns are mainly rectangular/sub-rectangular and ovate/obovate in plan view, while grinders have mostly spherical/discoidal and ovate/obovate forms. Grinding tools are predominantly made from different types of andesites (an extrusive volcanic rock), a material choice widely encountered in different communities with stone milling technology (e.g., [95–97]). Occasionally, grinders were also fashioned from quarzitic sandstone, orthoquartzite, metaquartzite and basalt. The tools sampled for this study derive from burnt and unburnt buildings attributed to the Middle and Late period of occupation (Fig 2 and Table 1).

Building 52 is located in the North Area and dates to 6700–6500 cal BC. This building was remodeled multiple times during its occupation history creating numerous activity areas, which included a long and narrow room situated to the west of the building (Sp. 91/92) that was used as a corridor in the final closure phase [98–100]. The floors within this room are

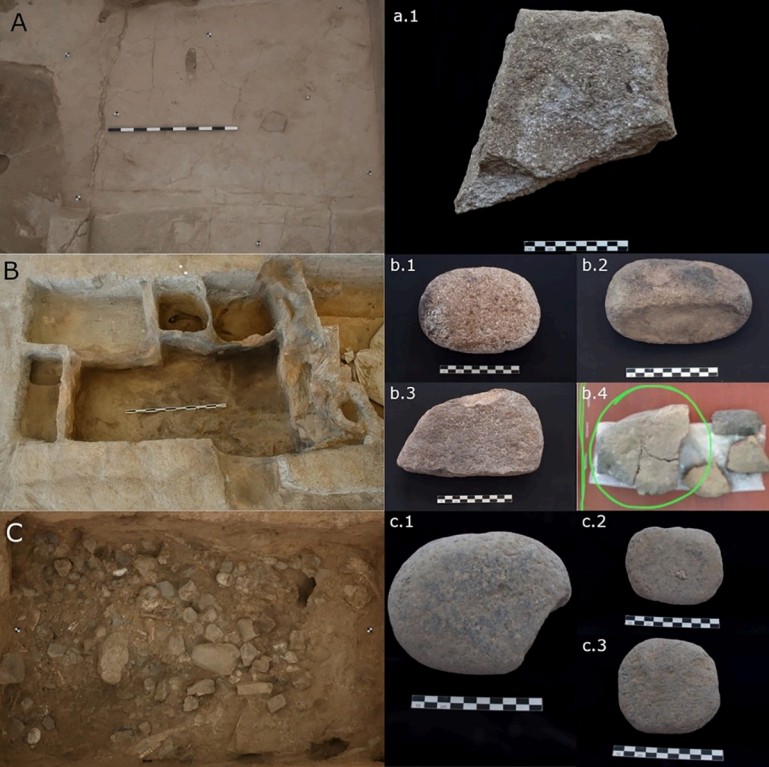

**Fig 2. Artefacts and buildings part of this study.** A) Fixed grinding installation, Building 80, Space 135; a.1). Quern 21767.x2; B) Storage room with clay bins Building 52, space 93; b.1) grinder 10292.x2; b.2) grinding/abrasive tool 10304.x6; b.3) grinding tool 10306.x11; b.4) grinding slab 10304.x8; C) Artefact cluster Building 44, unit 11648; c.1) grinder 11648.x14; c.2) grinder 11648.x8; c.3) grinder 11648.x22. Photo credits: Jason Quinlan, Kate Rose and Uğur Eyilik, Marco Madella (Çatalhöyük Research Project).

heavily burned and produced several artefacts, including a portable grinding tool (10306.x11) [98]. The burned artefact is made from medium-grained sandstone and has a slightly concave use-face. The object went through different episodes of use and modifications; it was originally larger and after breakage it was re-used in its present (smaller) form. In the northwest corner of the building there was a small storage room (Sp. 93) accessed from Space 91 via a small entrance (Fig 2B). This room had four clay bins arranged against the walls, the largest (F.2003) containing concentrations of naked barley grains and wild mustard, sheep/goat dung pellets and various stone artefacts among which a complete grinder made from porphyritic andesite (10292.x2) which was analyzed for this study [21, 80, 88, 98, 100]. The ovate and burned grinder has a slightly convex use-face and has been shaped by pecking. Traces of burning in the form of discoloration are encountered mainly on the margins of the tool, while the use-face of the tool does not exhibit the same type of alteration. The opposite surface and margins are for the most part covered with plaster residues. Upon the abandonment of the building the floor surface of Space 93 was strewn with botanical remains and artefacts, including a small-sized andesitic grinding slab (10304.x8) and an ovate grinding/abrading tool (10304.x6) [80, 88, 98, 100]. Artefact 10304.x6, which is made from a medium grained sandstone, survives complete and shows a moderate degree of burning. The implement has multiple use-faces, one of which is flat and two are convex. The tool was originally shaped by pecking but following its use there was no attempt to rejuvenate the flat use-face by re-pecking it. The botanical macro-remains associated with Building 52 suggest post-harvesting practices like dehusking and the

**Table 1. Grinding tools.**

| Building/ Space. | Object No. | Chronology cal BC. | Object Type. | Measurements mm. | Material/ Texture. | Preservation/ Condition. | Shape Plan/ Section. | Number & Shape of use-faces. | Microbotanical samples. |
|---|---|---|---|---|---|---|---|---|---|
| B.52/93 | 10292. x2 | 6700–6500 | Grinder. | 145.47 x 103.49 | Andesite/ Porphyritic. | Complete/Burnt-moderate. | Ovate/Ovate. | 1/Slightly convex. | 2 |
| B.52/93 | 10304. x6 | 6700–6500 | Grinding/ abrading tool. | 155.50 x 92.99 | Sandstone/ Medium grained, well cemented, well sorted. | Complete/Burnt-moderate. | Ovate/ Triangular. | 3/Flat & convex. | 4 |
| B.52/93 | 10304. x8 | 6700–6500 | Quern. | 228.00 x 148.00 | Andesite/ Porphyritic. | Almost complete (incomplete thickness)/Burnt-heavy. | Ovate/ Indeterminate. | 1/Concave. | 2 |
| B.52/91 | 10306. x11 | 6700–6500 | Grinding tool (small slab, originally larger lower grinding tool). | 190.00 x 108.41 | Sandstone/ Medium grained, well cemented, well sorted. | Complete/Burnt-moderate/heavy. | Irregular/ Irregular. | 1/Slightly concave. | 4 |
| B.80/135 | 21767. x2 | 6700–6500 | Quern/Fixed grinding installation. | 170.00 x 190.00 | Andesite/ Porphyritic. | Broken/Burnt-moderate. | Trapezoidal/ Irregular. | 1/Concave. | 4 |
| B.44/120 | 11648. x8 | 6500–6300 | Grinder. | 100.55 x 78.49 | Andesite/ Porphyritic. | Complete/Good. | Ovate/Plano-convex. | 1/Flat. | 2 |
| B.44/120 | 11648. x22 | 6500–6300 | Grinder. | 92.01 x 88.99 | Andesite/ Porphyritic. | Complete/Good. | Sub-square/ Lens. | 2 opposite/ Flat & slightly convex. | 4 |
| B.44/120 | 11648. x14 | 6500–6300 | Grinder. | 133.69 x 102.12 | Andesite/ Porphyritic. | Almost complete (small part of one end missing)/ Good. | Ovate/Plano-irregular. | 2 opposite/ Flat & convex. | 4 |

Description of grinding tools sampled for microbotanical and microwear analysis.

use of wild resources, such as almonds, peas and wild mustard [21, 80, 98, 101]. This assemblage related to plant storage suggests a diversified diet [21], and the spectrum of plant use in this building was probably even wider than what the macrobotanical evidence suggests [80]. All sampled tools come from the abandonment phase of the building (B.52.6).

Building 80 dated to 6700–6500 cal BC is located in the South Area and consists of two rooms: the main room (Space 135) to the north and a storage room (Space 373) to the south [102, 103]. Among the stone implements found in this building is a quern made from pink-coloured porphyritic andesite that was found embedded in the floor in the northern part of the main room (Sp.135) forming a fixed installation. It was placed in its fixed locale already broken in half during an advanced stage of the occupation of the building (building phase 2.3), and went out of use in building phase B.80.2.6 just before the building was destroyed by fire and was abandoned [90, 102, 103] (Fig 2A). The quern has one concave use-face and shows different episodes of use and maintenance. When the quern was plastered over at the end of its use-life, however, its use-face was in the process of being rejuvenated and part of the original surface of the use-face was removed by pecking. While the plastering of the tool partly protected the tool from the fire that destroyed the building the surface of the tool has been affected by the heat and for that reason part of the original surface has deteriorated. Macrobotanical remains recovered from the building in general and the areas around this artefact consisted

mainly of food processing debris such as cereal husks [81, 103]. The analysis of the fixed installation provides the opportunity to investigate plant processing activities directly associated with the actual occupation of the building (cf. [90]).

The remaining artefacts sampled for microbotanical analysis originated from Building 44, also located in the South Area and attributed to the Late period of occupation (6500–6300 cal BC) [104]. During the construction of the southwestern platform of the building, a rich deposit that included a large number of ground stone artefacts (n = 174), obsidian tools and debitage, faunal remains, bone tools, pottery sherds, stone beads, ochre nodules and figurines were deliberately placed in the foundation layer of this feature. (Fig 2C) [88, 104–108]. While previously this deposition was interpreted as the result of clearing/cleaning up activities of a household toolkit [20, 104], recent re-evaluation of the material suggests that this represents an intentional deposition that entailed the concerted actions of multiple social groups [108, 109]. Three complete grinders that are not burnt were sampled for microbotanical analysis (11648. x22; 11648.x8; 11648.x14). They are all made from grey-coloured porphyritic andesite and were operated with one hand. Grinder 11648.x8 has one flat use-face, while the other two grinders have one flat and one convex use-face on two opposite surfaces each. The use-faces of all three grinders were shaped by pecking prior to use. Although this sample represents a small portion of the ground stone tools placed in this feature, it provides an opportunity to study a depositional context where, contrary to buildings 80 and 52, ground stone artefacts are not directly related to the use-history of the building.

## 2.2 Sampling and laboratory procedures for microbotanical analysis

Microbotanical samples were obtained during the 2015 field season by members of the microbotanical team [110]. The tools sampled for residue analysis were selected in close collaboration with the ground stone team taking into account observed variability in the grinding tool assemblage in terms of raw materials, tool morphology and typology.

As standard practice, prior to 2012 ground stone artefacts at Çatalhöyük were not washed [23, 88], and during the final cycle of the excavations (2012–2017) a more rigorous procedure was followed with regard to the treatment objects kept for residue analysis received (e.g., minimal handling, storage of individual artefacts in new clean plastic bags, fast-track processing) [111]. The selected artefacts were transported in zipped plastic bags to a clean and isolated subset of the Çatalhöyük archaeobotany laboratory to proceed with the residue extraction. Each use-face of the artefacts was dry and wet brushed using distilled water, with a new toothbrush each time, to remove all residues trapped in the pits and crevices of the tool surfaces. A total of 29 residue samples were collected, including sediment samples used as controls, which were representative of the unit where the artefacts were found. The material recovered was stored in sterile lab tubes and exported from Turkey with permission from the Turkish Ministry of Culture to the Laboratory for Environmental Archaeology at Universitat Pompeu Fabra.

Various measures were adopted to avoid possible contamination during the field sampling and extraction procedures in the labs. At the field laboratory, all surfaces, walls and lab materials were cleaned with commercial bleach, no food was permitted in the area of the field laboratory while the extraction took place, and the door of the conditioned space was double sealed. No gloves were used and hands were washed with a soapy solution at the beginning of each sampling. The space was tested for contaminants at the end of each sampling cycle (trap slides). In the UPF lab, all surfaces and materials were exhaustively cleaned using a diluted solution of caustic soda (NaOH) and tested for contamination before performing the extractions (trap slides). The ventilation was cut and only one person was present during the extractions. Hairnets and masks were worn, and no gloves were used during the handling of the

samples to avoid any other source of contamination. The extraction procedure followed a combined protocol for phytoliths and starch extraction adapted from Pagan-Jimenez [59] and García-Granero et al. [112]. In this method, starch grains are first extracted from the sediments to avoid their exposure to any stronger chemical used for the phytolith extraction [113–115].

## 2.3 Microbotanical remains counting and registering

Samples were observed and recorded at 400x magnification using an Olympus BX51 microscope and Olympus Stream Basic software. Sample extracts were mounted in 50% glycerin and the whole suspensions were fully scanned. Starch grains were described using published literature and terminology (i.e., [116–118]). All starch grains found, including those broken but retaining diagnostic traits, were recorded, measured and allocated to type groups (i.e. [119]). Starch grains with extensive taphonomic damage were excluded from statistics since their morphological integrity was compromised, affecting any safe identification and interpretation [119–123]. When possible, starch identification was achieved using published data and the reference collection of the UPF Laboratory for Environmental Archaeology comprising more than 2,200 micro and macrobotanical specimens; the reference collection was also expanded with herbarium material (Botanical Institute of Barcelona) from wild geophyte species that are traditionally used in Turkey and the eastern Mediterranean.

Phytoliths were mounted with Entellan New® and scanned to count a minimum of 250 phytoliths when possible. Articulated phytoliths (silica skeletons) were recorded separately. Concentrations of phytoliths per gram of acid-insoluble fraction (AIF) were calculated following Albert et al. [124], and morphometric analyses were performed on complete cells and/or articulated forms following Ball [55] and Zhang et al. [125]. Descriptions followed the nomenclature of ICPN 1.0 [126] and 2.0 [127], although we maintained some names according to the ICPN 1.0 for the sake of clarity (following the suggestion of *nomina conservada* from ICPN 1.0), and identification was achieved using published material (e.g., [24, 54, 125, 128, 129]) and the Culture and Socio-Ecological Dynamics Research Group's reference collection.

## 2.4 Microwear analysis

Following the technological analysis of the ground stone assemblage a number of objects were selected for microwear analysis, including the set of tools that were further sampled for microbotanical analysis. The combination of microwear and microbotanical analyses discussed in this paper allowed not only for a better understanding of the plant exploitation strategies but also for a more nuanced appreciation of tool function at the site [89, 90]. The selection of material was guided by typological, technological and contextual criteria, as well as the overall condition of the material. While tools that survive in generally good condition without visible alterations were favoured, in some cases tools that show low or moderate heat alterations but derive from significant contexts (e.g., fixed grinding installation 21767.x2 from B.80) and/or survive intact were also included in the sampled material. In cases where the tools exhibited more intense burning, which affected the use-faces of the implements, these were not subjected to microwear analysis.

Microwear analysis was conducted at low (up to 100x) and high magnifications (100x and 200x) utilising a stereomicroscope (Leica M80) with an external, oblique light source and with a coaxial illumination unit (Leica M80 LED5000 CXI, magnifications up to 230x), and an incident light microscope (Nikon Eclipse LV100NPOL). In addition to the on-site microwear analysis of the tools, silicon casts (Affinis® Light Body Coltene and Provil® Novo Light Regular Set Heraeus Kulzer), which can accurately replicate the micro-topography of the tool surface, were taken. Recorded wear patterns included grain edge rounding, levelling, grain

extraction, micropolish features, texture and development, microstriations, microfractures, and the presence of residues [33, 130]. The microwear patterns observed on the archaeological tools were interpreted in relation to the reference collection of experimentally used tools housed at the Laboratory for Material Culture Studies at Leiden University as well as published data (e.g., [31, 45, 130, 131]). The Leiden reference collection includes more than 100 stone tools that were used singly or in pairs to process different plant, mineral and animal materials including cereals, acorns, legumes, oilseeds, wood, flint and other stones, ochre, clay, bone, antler, shell, and hide. The experiments were executed on average for 180 minutes using manual approaches and motions included crushing, grinding, abrading, polishing and softening. The experiments were performed mainly on quartzitic sandstone, but granite, basalt and quartzite were also used (see also [35, 132–134]) (Fig 3).

Microwear analysis of grinding tools has shown that broad classes of worked materials (plant, mineral, animal) are associated with distinctive types of wear patterns including characteristic types of micropolish that develop on the surface of the tools during different activities [135, 136]. Recent research has highlighted variations in microwear signatures that result from the processing of different plant categories (e.g., cereals, legumes, nuts and seeds), variations that can also be linked to the state (dry, wet) and the type of processing of the processed material (soaking, roasting) (e.g., [122]). No well-established use-wear signatures can be securely associated with the processing of USOs due to the lack or limited number of experiments focusing on USOs [137]. While the identification of diagnostic use-wear signatures associated with the processing of different plant materials has been achieved at an experimental level [137, 138], it is not always possible to apply this detail of interpretation to the archaeological materials. This is particularly the case for archaeological tools that have been used against multiple contact materials and therefore display mixed wear signatures. Experiments tend to focus on the processing of a single type of worked material and there is currently little detailed evaluation on the development of use-wear traces on tool surfaces used to process more than one material (cf. [131]). Moreover, even when restricting to plant materials, interpretations can be hindered in the case of tools that were used to process a diverse assemblage of plants or plant parts on the same tool surfaces: cereal grinding results in well-developed wear traces at a faster rate than other types of plant processing therefore concealing–potentially underdeveloped -use-wear traces from other types of plants such as legumes, USOs, softer plants such as herbs and grasses [137]. Hence, the analysis of microbotanical residues has the potential to provide a wider and more detailed identifications on the plant assemblage and materials that were processed on grinding tools (cf. [136]).

## 3.0 Results

### 3.1 Starch grains

A total of 567 starch grains were recorded. Typologies were first created when at least two starch grains with shared morphologies were found during the scanning. Once all samples were scanned, these earlier identifications were refined and grouped into a total of 14 different types (Fig 4A and 4B). A considerable number of starch grains presented heavy taphonomic damage and for that reason were not included in the size range statistics.

**Type 1**–184 (R: 35.08–10.99 μm, Av: 21.1063 μm, SD: 5.04102 μm, n = 30). Round or slightly oval on plane view and elliptical when rotated, occasionally presenting a distinctive surface feature resembling circular hollowness or craters (Fig 4A, a). This type commonly occurs in the *Pooideae* sub-family and is referred in the literature as *Triticeae* Type A [139–142].

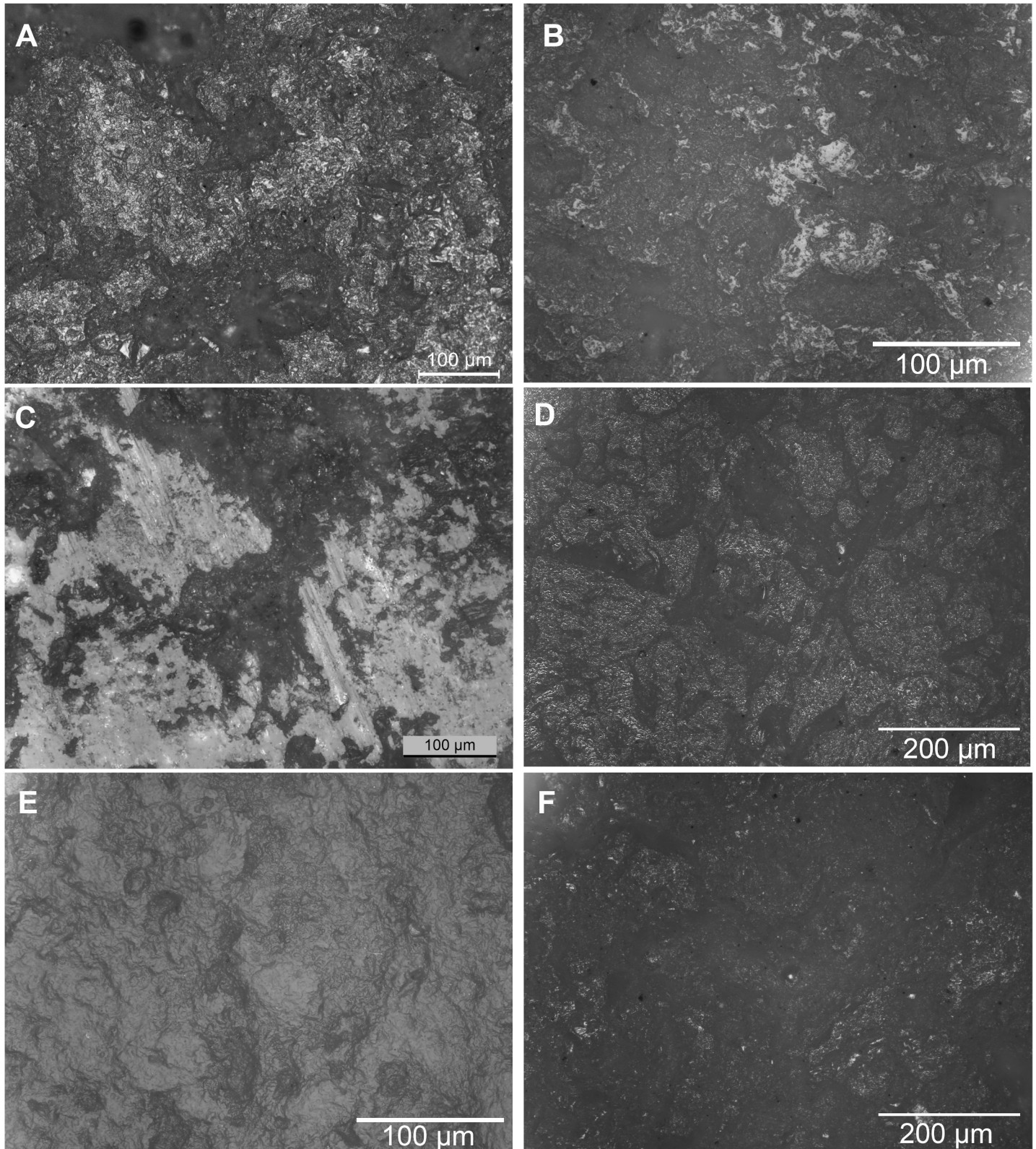

**Fig 3. Microwear traces on experimental tools.** A) grinding acorns (200x); B) abrading wood (200x); C) grinding basalt (200x); D) grinding cereals (einkorn wheat) (100x); E) burnishing/polishing leatherhard clay (200x); F) grinding legumes (lentils) (100x). Courtesy of Leiden Laboratory for Material Culture Studies.

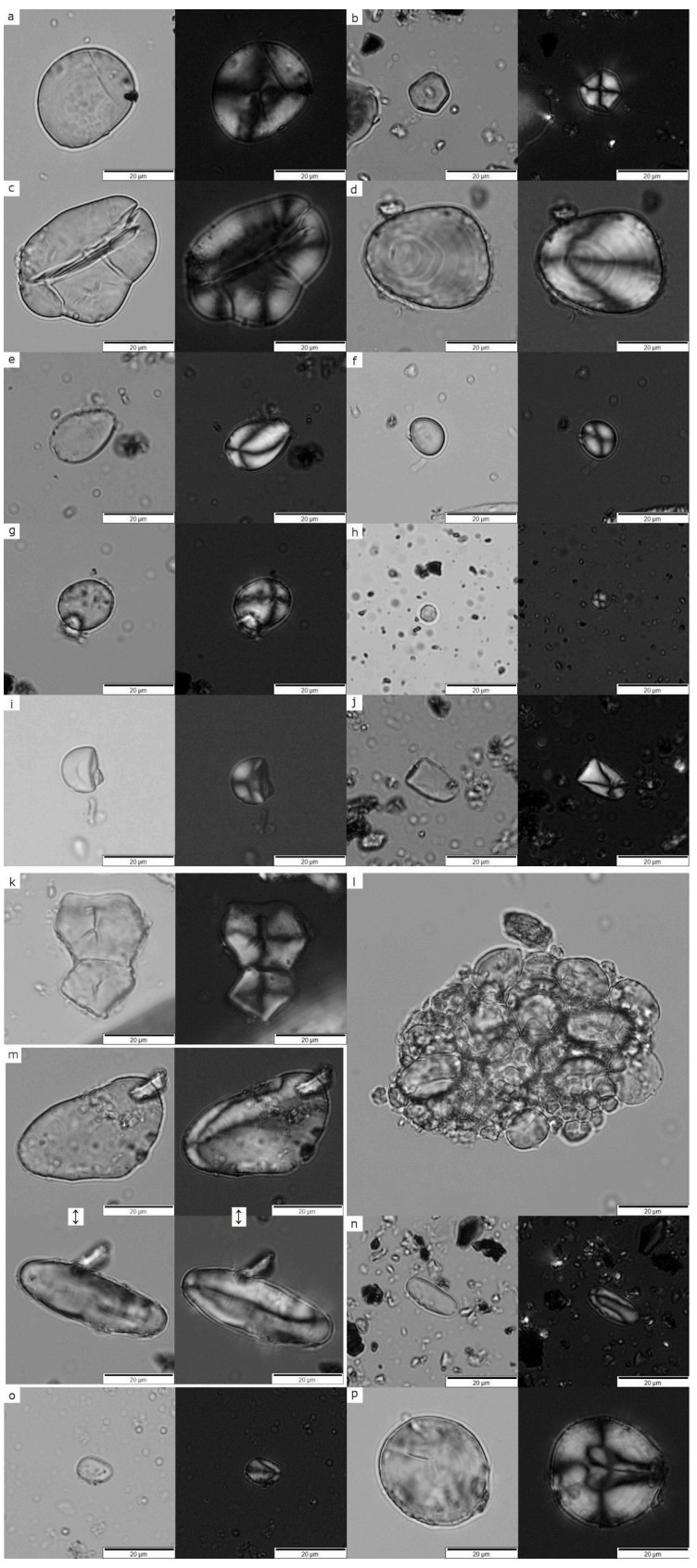

**Fig 4.** A. Archaeological starch grains recovered from ground tools under bright and cross-polarized light at 400x magnification. a) Type 1, Triticeae (*type A*) starch grain exhibiting crater-like surface; b) Type 2, polyhedral starch grain from Panicoideae with open hilum; c) Type 3, Faboideae starch grain with longitudinal fissure; d) Type 4, big USO starch grain with eccentric cross; e) Type 5, small conoid USO starch grain, notice the hilum location on the wider end of the grain; f) Type 6, small round to oval shape starch grain; g) Type 7, spherical/oval starch grain with an eccentric cross probably from USOs; h) Type 8, Small round starch grains with a visible cross, probably Triticeae (*type B*); i) Type 9, hemispherical starch grain, unidentified; j) Type 10, elongated plano-convex shape (bell shape) in plain view exhibiting an eccentric cross, unidentified. B. Archaeological starch grains recovered from ground tools under bright and cross-polarized light at 400x magnification. k) Type 11, two big polyhedral starch grains, probably from a cluster; l) Cluster of Triticeae starch grains from type 1 and 8; m) Type 12, Big USO starch grain, triangular elongated on plain view and elliptical when rotated (below); n) Type 13; small cylindrical starch from USOs exhibiting an eccentric cross; o) Type 14; small conoid starch grain from USOs; p) Unidentified compound starch grain.

**Type 2**–45 (R: 16.19–10.95 μm, Av: 12.8243 μm, SD: 1.41710 μm n = 16). Medium size starches with an angular and polyhedral to sub-round shape in plain view and angular phases when rotated (Fig 4A, b). It has a centric hilum and an extinction cross with straight arms. This type is common to the *Panicoideae* subfamily [45, 128, 141, 143, 144].

**Type 3**–48 (R: 28.42–15.48 μm, Av: 21.23 μm, SD: 5.0534 μm n = 8). Commonly oval, truncated or reniform with visible lamellae (Fig 4A, c). The extinction cross can be asymmetric in some cases and it is common to see a cleft or deep longitudinal fissure on the centric area splitting the grain into two or more fragments. This group is commonly found in legumes and ascribed to the *Fabeae* tribe [116, 119].

**Type 4**–3 (R: 38.16–47.78 μm, Av: 42.97 μm, SD: 4.81μm n = 2). The form is triangular/elongated or ovoid shape. It has an extremely eccentric hilum producing a cross with curved to wavy arms (Fig 4A, d). The lamellae is visible, forming eccentric rings. This type is found in USOs [117, 118, 143, 145].

**Type 5**–3 (R: 20.91–11.19 μm, Av: 14.5133 μm, SD: 4.524 μm). Presenting conoid shapes with a distinctive eccentric hilum locate on the distal end of the grain with an extinction cross with curved to wavy arms (Fig 4A, e). This type is commonly found in USOs [143, 145].

**Type 6**–38 (R: 15.42–5.45 μm, Av: 10.045 μm, SD: 2.897 μm n = 18). Some grains of this group present a round to oval shape with an occasional central depression or ¨Y¨ shape fissure in the mesial area and a visible extinction cross (Fig 4A, f). The size range and morphological characteristics of this type are similar to those reported on *Typha spp.* and *Cyperaceae spp.* [143, 146–148]. (Further discussion below).

**Type 7**–3 (R: 27.00–16.64 μm, Av: 21.233 μm, SD: 4.3101 μm). Circular to oval in plain view and spherical when rotated with a distinct eccentric hilum and visible lamellae forming concentric circles and an extinction cross with straight arms (Fig 4A, g). This form is similar to examples found geophytes presented on various publications and our reference collection [143, 145, 149]. However, its shape is not exclusive to a single taxon.

**Type 8**–15 (R: 6.65–4.80 μm, Av: 5.5833 μm, SD: 0.7813 μm n = 3). Small round starch grains. The extinction cross is not always visible but, when present, it forms a cross with straight arms (Fig 4A, h). These could be transient starch grains that are usually not diagnostic [150]. However, this type is also found in clusters of *Triticeae* Type A, both in our reference collection and in the archaeological samples. Then, it is more likely that this group is represented by *Triticeae* Type B [42, 142, 151].

**Type 9**–5 (R: 17.24–12.76 μm, Av: 15.054 μm, SD: 1.739 μm). Globular in plain view, and dome/hemispherical faceted when rotated (Fig 4A, i). It has a centric hilum and exhibits a distinctive extinction cross. This type remains unidentified.

**Type 10**–4 (R: 17.13–14.54 μm, Av: 16.22 μm, SD: 1.5840 μm n = 2). Elongated plano-convex shape (bell shape) with one round point and a distal flat base in plain view and cylindrical

when rotated (Fig 4A, j). It presents an eccentric extinction cross under polarized light closer to the proximal area with occasionally wavy arms. (further discussion below).

**Type 11**–130 (R: 25.58–15.15 μm, Av: 19.2548 μm, SD: 2.9068 μm, n = 69). Big, polyhedral starch grains with centric hilum and cross (Fig 4B, k). This type is commonly associated with the big *Panicoideae* taxa (i.e., [40, 44, 46]).

**Type 12**–1 (46.40 μm). Triangular/elongated in plain view and elliptical when rotated with an extremely eccentric hilum, producing a cross with curved to wavy arms and eccentric rings (Fig 4B, m). This type is found in USOs [117, 118, 143].

**Type 13**–1 (16.40 μm). Cylindrical elongated shape with extremely eccentric hilum and an extinction cross with curved to wavy arms (Fig 4B, n). Based on reference collections and in accordance with the literature, starches from this group are produced by different geophyte taxa such as the *Iridaceae* [40, 117, 143, 145, 152, 153].

**Type 14**–2 (R: 10.45–9.76 μm, Av: 10.105 μm, SD: 0.4879 μm). Conoid to ovoid shapes with extremely eccentric hilum and an extinction cross with curved to wavy arms similar to type 5 but the hilum is located on the most angular end (Fig 4B, o). This shape is commonly found in USOs.

**Not defined type**—This group contains 85 grains mostly heavily broken or gelatinized that hamper their identification into one of the above groups.

**3.1.1 Taxonomic groups and ID.** Starch grains were generally ubiquitous, diverse and abundant in all examined artefacts, with the exception of the quern in Building 80 (Table 2). The 14 types identified during the slide scanning have been grouped into four broad taxonomic groups (Fig 5).

*3.1.1.1 Triticeae.* This group includes starch types 1 and 8 probably representing one or more of the cereal species documented at the site (e.g., ¨new type¨ wheat, *T. dicoccum*, *T. monococcum*, *T. aestivum* and *H. vulgare* ¨two and six-row¨) [80, 81]. The analyses also produced 7 starch aggregates (i.e., clusters) composed of *Triticeae* types 1 and 8, some with more than 50 single grains (Fig 4B, l). These were not counted since it was not possible to register every single starch grain within the clusters.

*3.1.1.2 Panicoideae.* This group includes type 2 and type 11 with polyhedral shapes. Polyhedral starch grains occur when tightly packed together in the amyloplast [37]. There are various genera of this sub-family occurring in Turkey that can be producing these polyhedral starches: *Echinochloa* (*E. colona*, *E. crus-galli*), *Setaria* (*S. verticillata*, *S. viridis*), *Sorghum* (*S. halepense*) and *Panicum* (*P. repens*) [154]. Starch size, shape and features have been parameters previously used to differentiate and identify these species in archaeological contexts [45, 128]. Among the mentioned species *S. verticillata* is known to produce medium-size starch grains reaching up to 15 μm [45, 128], making it a possible candidate for type 2. On the other hand, we notice that *S. halepense* produces big starch grains (R: 27.91–12.49 μm, Av: 18.9849 μm, SD: 2.8379 μm, n = 150) matching in size with type 11, therefore emerging as a possible candidate for this type. Nevertheless, big polyhedral starch grains are not exclusive of Panicoids (i.e., [116–118, 155, 156], we keep type 11 identification as tentative.

*3.1.1.3 Fabeae.*—Represented by starch type 3, probably originating from one or more of the pulse species found as seeds at the site such as *Pisum sativum*, *Lens culinaris*, *Vicia ervilia*, *Cicer arietinum* [80, 81].

*3.1.1.4 Underground Storage Organs.* This group is formed by 7 types (type 4–7, 12–14). Based on our reference collection as well as published data this group includes species from different monocot plants such as the *Iridaceae* family represented by type 13 [40]. However, it is not possible to attain a lower taxonomic identification because different species in this family produce starch grains with similar morphometric characteristics (i.e., [117, 118]. A detailed morphometric analysis from several species of this family needs to be done to garner high

**Table 2. Starch count.**

| Artefact | 10292.x2 | | 10304.x6 | | | | 10304.x8 | | | 10306.x11 | | | 21767.x2 | | | | 11648.x8 | | | 11648.x22 | | | 11648.x14 | | | | Control | | |
|---|---|---|---|---|---|---|---|---|---|---|---|---|---|---|---|---|---|---|---|---|---|---|---|---|---|---|---|---|---|
| Origin | Dry | Wet | F1-Dry | F2-Dry | F1-Wet | F2-Wet | Dry | Wet | F1-Dry | F2-Dry | F1-Wet | F2-Wet | F1-Dry | F2-Dry | F1-Wet | F2-Wet | Dry | Wet | F1-Dry | F1-Wet | F2-Dry | F2-Wet | F1-Dry | F1-Wet | F2-Dry | F2-Wet | B.52 | B.80 | B.44 |
| Sample g. | 0.1008 | 0.1671 | 0.0027 | 0.0085 | 0.0238 | 0.0232 | 0.0706 | 0.1339 | 0.0061 | 0.0087 | 0.1203 | 0.0194 | 1.2502 | 1.5109 | 0.4819 | 0.9789 | 0.0137 | 0.01299 | 0.0014 | 0.024 | 0.0107 | 0.0914 | 0.1582 | 0.1900 | 0.0418 | 0.0614 | 6.1878 | 3.2638 | 1.0521 |
| Type 1 | 22 | 0 | 12 | 7 | 5 | 7 | 25 | 0 | 6 | 56 | 1 | 7 | 0 | 0 | 1 | 1 | 0 | 0 | 13 | 0 | 2 | 6 | 7 | 1 | 0 | 4 | 0 | 0 | 1 |
| Type 2 | 2 | 2 | 0 | 1 | 1 | 4 | 3 | 0 | 0 | 4 | 5 | 9 | 0 | 0 | 0 | 0 | 0 | 0 | 1 | 0 | 3 | 2 | 2 | 0 | 2 | 2 | 0 | 0 | 2 |
| Type 3 | 10 | 0 | 2 | 4 | 1 | 3 | 3 | 0 | 0 | 8 | 0 | 1 | 0 | 3 | 0 | 0 | 7 | 0 | 0 | 0 | 1 | 0 | 2 | 0 | 0 | 3 | 0 | 0 | 0 |
| Type 4 | 0 | 0 | 0 | 0 | 0 | 0 | 1 | 0 | 0 | 1 | 0 | 0 | 0 | 0 | 0 | 0 | 0 | 0 | 0 | 0 | 0 | 0 | 0 | 0 | 0 | 1 | 0 | 0 | 0 |
| Type 5 | 0 | 1 | 0 | 0 | 0 | 0 | 0 | 0 | 0 | 1 | 1 | 0 | 0 | 0 | 0 | 0 | 0 | 0 | 0 | 0 | 0 | 0 | 0 | 0 | 0 | 0 | 0 | 0 | 0 |
| Type 6 | 1 | 0 | 6 | 0 | 0 | 1 | 5 | 0 | 0 | 9 | 1 | 6 | 1 | 0 | 0 | 0 | 0 | 1 | 1 | 0 | 0 | 2 | 2 | 0 | 0 | 2 | 0 | 0 | 0 |
| Type 7 | 1 | 0 | 1 | 0 | 0 | 0 | 0 | 0 | 1 | 0 | 0 | 0 | 0 | 0 | 0 | 0 | 0 | 0 | 0 | 0 | 0 | 0 | 0 | 0 | 0 | 0 | 0 | 0 | 0 |
| Type 8 | 0 | 0 | 0 | 0 | 1 | 0 | 0 | 0 | 0 | 6 | 0 | 7 | 0 | 0 | 0 | 0 | 0 | 0 | 2 | 0 | 0 | 0 | 0 | 0 | 0 | 1 | 0 | 0 | 0 |
| Type 9 | 0 | 2 | 0 | 1 | 1 | 0 | 0 | 0 | 0 | 0 | 0 | 0 | 0 | 0 | 0 | 0 | 0 | 0 | 0 | 0 | 0 | 0 | 0 | 0 | 0 | 0 | 0 | 0 | 0 |
| Type 10 | 1 | 0 | 0 | 0 | 0 | 0 | 2 | 0 | 0 | 0 | 1 | 0 | 0 | 0 | 0 | 0 | 0 | 0 | 0 | 0 | 1 | 0 | 0 | 0 | 0 | 0 | 0 | 0 | 0 |
| Type 11 | 8 | 4 | 3 | 6 | 1 | 9 | 14 | 1 | 4 | 23 | 8 | 18 | 3 | 3 | 1 | 0 | 1 | 6 | 2 | 0 | 1 | 4 | 8 | 0 | 0 | 1 | 0 | 0 | 4 |
| Type 12 | 0 | 0 | 1 | 0 | 0 | 0 | 0 | 0 | 0 | 0 | 0 | 0 | 0 | 0 | 0 | 0 | 0 | 0 | 0 | 0 | 0 | 0 | 0 | 0 | 0 | 0 | 0 | 0 | 0 |
| Type 13 | 0 | 0 | 0 | 0 | 0 | 0 | 1 | 0 | 0 | 0 | 0 | 0 | 0 | 0 | 0 | 0 | 0 | 0 | 0 | 0 | 0 | 0 | 0 | 0 | 0 | 0 | 0 | 0 | 0 |
| Type 14 | 0 | 0 | 0 | 0 | 0 | 0 | 0 | 0 | 0 | 0 | 0 | 2 | 0 | 0 | 0 | 0 | 0 | 0 | 0 | 0 | 0 | 0 | 0 | 0 | 0 | 0 | 0 | 0 | 0 |
| No Type | 7 | 0 | 3 | 2 | 0 | 4 | 6 | 0 | 0 | 29 | 3 | 11 | 0 | 0 | 1 | 0 | 0 | 2 | 2 | 0 | 4 | 2 | 1 | 0 | 2 | 4 | 0 | 0 | 2 |
| Aggregates >50 | 2 | 0 | 0 | 1 | 0 | 0 | 0 | 0 | 0 | 0 | 0 | 0 | 0 | 0 | 0 | 0 | 0 | 0 | 1 | 0 | 0 | 0 | 0 | 0 | 0 | 0 | 0 | 0 | 0 |
| Aggregates <50 | 1 | 0 | 0 | 0 | 1 | 0 | 0 | 0 | 0 | 0 | 0 | 0 | 0 | 0 | 0 | 0 | 8 | 0 | 1 | 0 | 0 | 0 | 0 | 0 | 0 | 0 | 0 | 0 | 0 |
| Total | 52 | 9 | 28 | 21 | 9 | 28 | 60 | 1 | 11 | 137 | 20 | 61 | 0 | 7 | 3 | 1 | 8 | 9 | 21 | 0 | 11 | 16 | 22 | 1 | 4 | 18 | 0 | 0 | 9 |
| Triticeae | 22 | 0 | 12 | 7 | 5 | 7 | 25 | 0 | 6 | 62 | 1 | 14 | 0 | 0 | 1 | 1 | 0 | 0 | 15 | 0 | 2 | 6 | 7 | 1 | 4 | 4 | 0 | 0 | 1 |
| Panicoideae | 2 | 2 | 0 | 1 | 1 | 4 | 3 | 0 | 0 | 4 | 5 | 9 | 0 | 0 | 0 | 0 | 0 | 0 | 1 | 0 | 3 | 2 | 2 | 0 | 2 | 2 | 0 | 0 | 2 |
| Fabeae | 10 | 0 | 2 | 4 | 1 | 3 | 3 | 0 | 0 | 8 | 5 | 1 | 0 | 3 | 0 | 0 | 7 | 0 | 0 | 0 | 1 | 0 | 2 | 0 | 0 | 3 | 0 | 0 | 0 |
| USO | 1 | 1 | 7 | 0 | 0 | 1 | 7 | 0 | 1 | 11 | 2 | 8 | 0 | 1 | 0 | 0 | 0 | 1 | 1 | 0 | 0 | 2 | 2 | 0 | 0 | 3 | 0 | 0 | 0 |

Results, provenance, and broader taxonomic groups recorded on each sample.

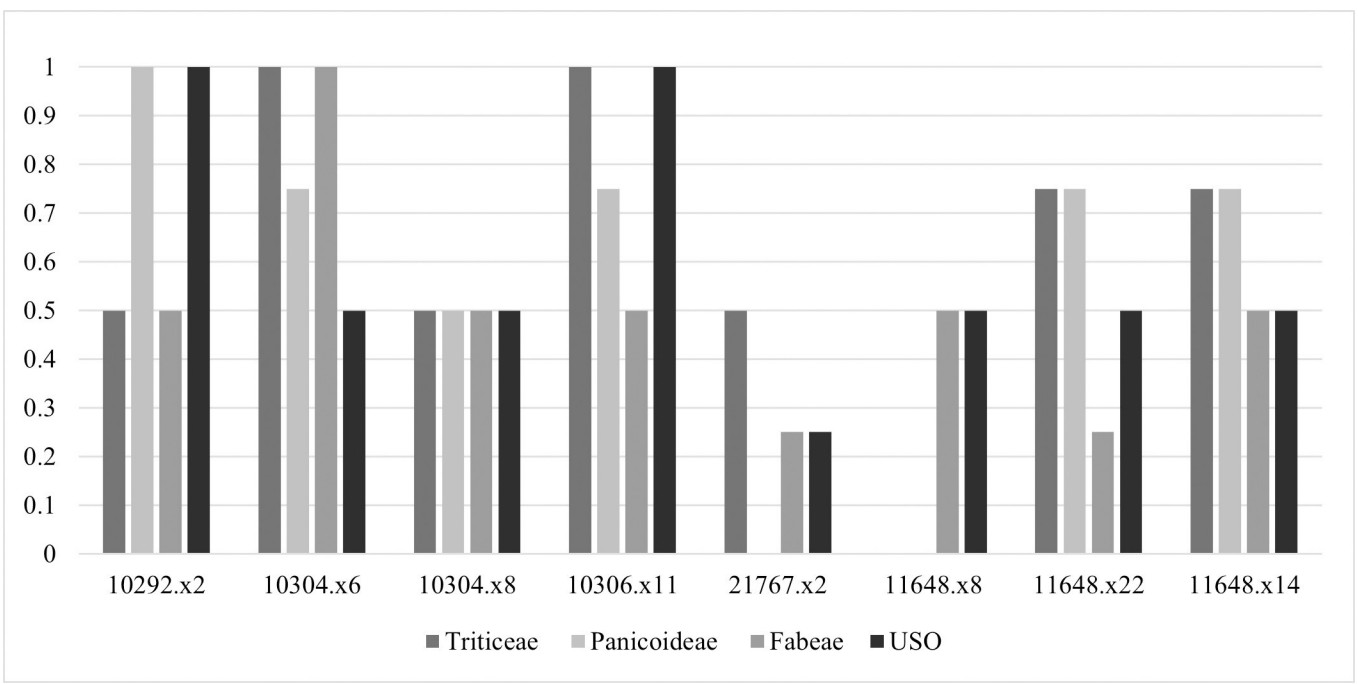

**Fig 5. Starch ubiquity by artefact.** Based on the presence and absence of starch taxonomic groups on each artefact.

resolution data that could reveal specific diagnostic features at different taxonomic levels. The same occurs with the pyriform/conoid forms of types 4 and 12, similar to starch produced by geophytes found in the *Liliaceae* family [117]. Another geophyte with all probability present in the USOs is the *Cyperaceae* family (type 6). Contrary to *Iridaceae* and *Liliaceae*, which are not represented in the macrobotanical assemblage, there are remains of *Cyperaceae* underground organs at the site [80]. In the case of *Cyperaceae* starch grains, lower taxonomic identification is difficult due to starch grain attributes shared within the family [118]. This makes morphometrics alone unsuitable or difficult for the identification of starches from this family in the archaeological record (i.e., [93, 157, 158]). Furthermore, some type 6 starches have similarities with starch grains from cattail (*Thypa* spp.) (i.e. [146]), a trait previously noticed by other researchers at the site [93]. Therefore, a separation between *Cyperaceae* and *Thypa* is not suggested, and both taxa might have been represented at the site. Other USO types, such as 5, 7 and 14, currently remain unidentified.

*3.1.1.5 Not identified.* The remaining types presented lower taxonomic value and it was not possible to assign them to any specific taxonomic group and therefore we have excluded these findings from further discussions. Type 10 for example resembles some forms found in geophytes. However, a study by Aceituno Bocanegra and López Sáez [159] on starch characterization from the genus *Triticum* and *Hordeum* noticed the occurrence of this morphotype too. This was confirmed with samples from our reference collection, which also showed its presence in *Secale* sp. More research must be done before considering the taxonomic value of this type. Finally, type 9 is commonly found in starch clusters of two or more grains occurring in seeds and USOs and therefore it does not have much taxonomic significance at the moment [117, 143, 149, 160].

**3.1.2 Taphonomy and contamination.** Control sediment samples were rich in phytoliths and devoid of starch except for Building 44 where the sediment produced 9 heavily damaged starch grains. The presence of starch grains in peripheral sediments has been broadly discussed

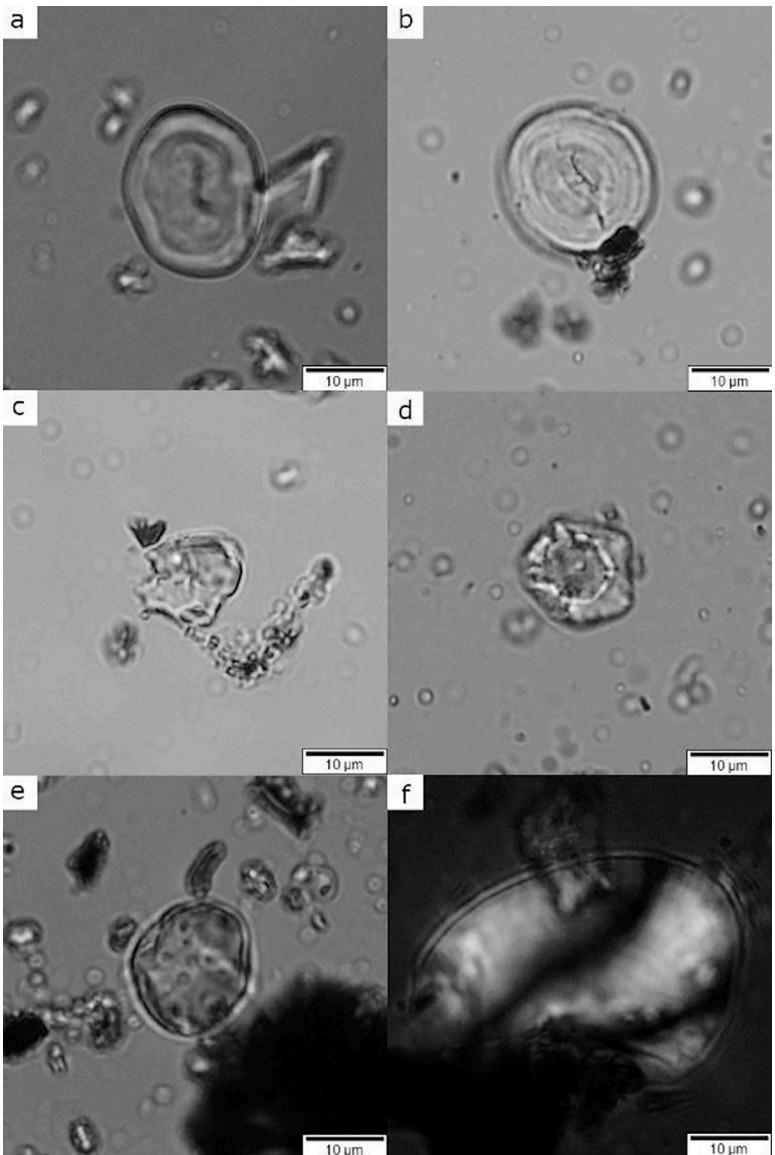

**Fig 6. Archaeological starch grains recovered from grinding stones with evident structural damage caused by anthropic, biological and taphonomic processes, observed under 400x magnification.** a–b) Type 1 starch grains with signs of initial stages of gelatinization; c) Type 2 starch grain presenting signs of initial stages of gelatinization; d) Type 2 starch grain with a central depression, pitting and channeling probably caused by enzymatic damage; e) Unidentified starch grain exhibiting pitted surface commonly caused by enzymes and microbial activity; f) Broken USO starch grain, probably caused by mechanical action (grinding, pounding, or mashing). Damage identifications based on Henry et al. [119]; Wang et al. [172]; Pagán-Jiménez [166, 171].

and, despite airborne contamination during fieldwork can always be a threat (i.e., [161]), multiple studies and experiments clearly suggest that starch recovered from sediments surrounding the analyzed artefact is most probably the result of a transfer from the artefact itself, or originating from in-situ activities [114, 115, 162–165].

Most of the starch grains from Çatalhöyük had some form of structural and surface damage (Fig 6). Damage on starch grains has long been discussed and there are numerous studies on the effects resulting from human activities such as boiling, parching, pounding and fermenting (i.e., [40, 119, 120, 122, 123, 152, 153, 166–169]). The secure identification of the type of

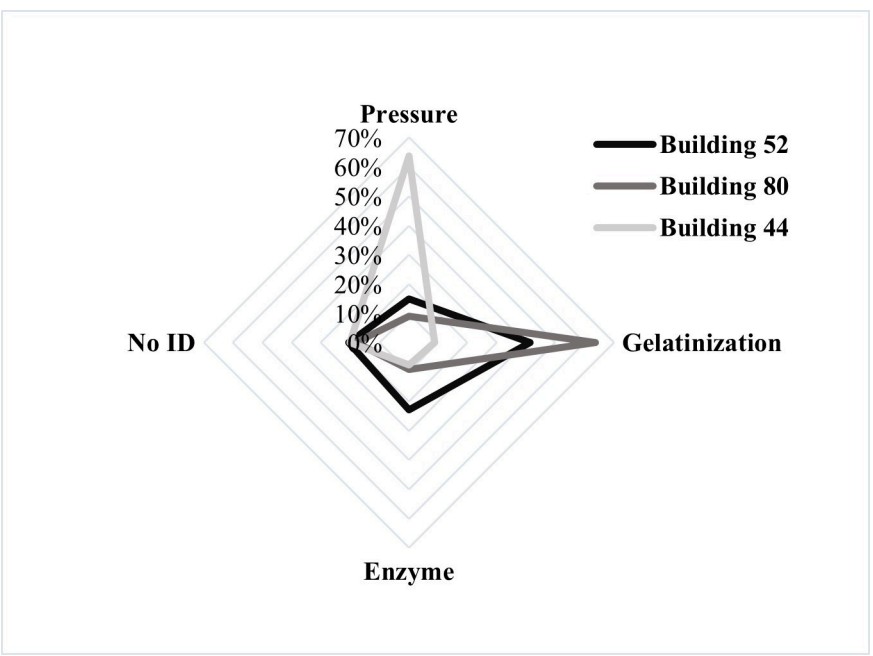

**Fig 7. Starch damage.** Proportions and distribution of starch damage by building.

damage in archaeological starch grains can be complex and relies heavily on context [170], but also in strong and reliable supporting data coming from experimental research [123, 166–169, 171, 172]. We noticed a predominance of starch grains with signs of gelatinization at various developing stages from artefacts in Buildings 80 and 52 (Fig 7). Full gelatinization may occur due to different causes, most commonly when starch is exposed to temperatures above 40˚C in the presence of water or in very high humidity [123, 160, 166, 171, 173, 174]. However, gelatinization-like damage was much less common in starch from Building 44 as this building was not burnt, as was the case for buildings 52 and 80, and the artefacts were sealed soon after deposition. Therefore, gelatinization-like damage in our samples in all probability is not related to intentional food processing activities or cooking, but to post-depositional processes such as the artefact's exposure to heat when both buildings caught fire. These observations are useful as they confirm that the recovered starches are of secure provenance and not the product of modern contamination.

## 3.2 Phytolith morphotypes and morphometric results

Phytoliths were recovered and are abundant on most sampled artefacts, except for both grinders from Building 52 and a grinder from Building 44 (11648.x8), and only artefacts 21767.x2 from Building 80, and 11648.x22 and 11648.x14 from Building 44 reached or surpassed the 250 phytoliths count (Table 3). The most abundant morphotypes found are long cells from the *Poaceae* family such as elongated dendritic and echinates commonly found in cereals and other grasses inflorescences (Fig 8). Phytoliths from *Phragmites* leaves and culm and papillae cells from *Cyperaceae* (sedges) were also observed. Other forms such as globular echinates (palms) were identified on the quern from Building 80 and a grinder in Building 44 (11648. x22). This last artefact also produced globular psilate morphotypes from Eudicot leaves' mesophyll. (Fig 9). Overall, the phytoliths assemblages from artefacts 11648.x22 and 11648.x14 were very similar in terms of morphotype diversity, in contrast to artefact 21767.x2 showing a high presence of cells from cereals inflorescence.

**Table 3. Phytoliths count.**

| Artefact | 10292.x2 | | 10304.x6 | | | | 10304.x8 | | 10306.x11 | | | | 21767.x2 | | | | 11648.x8 | | | | 11648.x22 | | 11648.x14 | | | | Control | | |
|---|---|---|---|---|---|---|---|---|---|---|---|---|---|---|---|---|---|---|---|---|---|---|---|---|---|---|---|---|---|
| Origin | Dry | Wet | F1-Dry | F2-Dry | F1-Wet | F2-Wet | Dry | Wet | F2-Dry | F1-Dry | F1-Wet | F2-Wet | F1-Dry | F2-Dry | F1-Wet | F2-Wet | Dry | Wet | F1-Dry | F1-Wet | F2-Dry | F2-Wet | F1-Dry | F1-Wet | F2-Dry | F2-Wet | B.52 | B.80 | B.44 |
| Sample Weight g | 0.1008 | 0.1671 | 0.0027 | 0.0085 | 0.0238 | 0.0232 | 0.0706 | 0.1339 | 0.0087 | 0.0061 | 0.1203 | 0.0194 | 1.2502 | 1.5109 | 0.4819 | 0.9789 | 0.0137 | 0.013 | 0.0014 | 0.024 | 0.0107 | 0.0914 | 0.1582 | 0.1900 | 0.0418 | 0.0614 | 6.1878 | 3.2638 | 1.0521 |
| Phytoliths x g | 0 | 10,773 | 5,072 | 0 | 0 | 943 | 2,956 | 2,049 | 846 | 0 | 9,449 | 0 | 134,851 | 367,840 | 379,553 | 802,671 | 0 | 0 | 0 | 3,154 | 0 | 21,914 | 12,038 | 16,641 | 16,406 | 37,016 | 61 | 86,823 | 1,301 |
| Echinates | 0 | 1 | 2 | 0 | 0 | 1 | 15 | 3 | 0 | 0 | 6 | 0 | 9 | 16 | 16 | 14 | 0 | 0 | 0 | 0 | 0 | 21 | 7 | 15 | 6 | 12 | 4 | 16 | 18 |
| Dendritics | 0 | 1 | 3 | 0 | 0 | 2 | 12 | 1 | 0 | 0 | 4 | 0 | 71 | 82 | 95 | 63 | 0 | 0 | 0 | 1 | 0 | 25 | 29 | 62 | 10 | 16 | 4 | 95 | 14 |
| Long Smooth | 0 | 3 | 0 | 0 | 0 | 0 | 5 | 0 | 0 | 0 | 4 | 0 | 23 | 8 | 13 | 17 | 0 | 0 | 0 | 3 | 0 | 5 | 4 | 11 | 4 | 5 | 1 | 20 | 1 |
| Long non-Smooth | 0 | 0 | 0 | 0 | 0 | 5 | 9 | 3 | 1 | 0 | 4 | 0 | 11 | 4 | 5 | 1 | 0 | 0 | 0 | 8 | 0 | 8 | 3 | 9 | 3 | 2 | 2 | 2 | 11 |
| Undulate | 0 | 0 | 0 | 0 | 0 | 0 | 0 | 0 | 0 | 0 | 0 | 0 | 6 | 0 | 5 | 1 | 0 | 0 | 0 | 4 | 0 | 0 | 0 | 4 | 0 | 0 | 0 | 1 | 0 |
| Bilobate | 0 | 0 | 0 | 0 | 0 | 0 | 0 | 0 | 0 | 0 | 0 | 0 | 2 | 0 | 0 | 4 | 0 | 0 | 0 | 1 | 0 | 1 | 2 | 0 | 1 | 7 | 0 | 1 | 1 |
| Polylobate | 0 | 0 | 0 | 0 | 0 | 0 | 0 | 0 | 0 | 0 | 0 | 0 | 0 | 1 | 0 | 0 | 0 | 0 | 15 | 0 | 0 | 0 | 1 | 0 | 0 | 4 | 0 | 3 | 0 |
| Cross | 0 | 0 | 0 | 0 | 0 | 0 | 0 | 0 | 0 | 0 | 0 | 0 | 0 | 0 | 0 | 0 | 0 | 0 | 0 | 1 | 0 | 0 | 2 | 1 | 0 | 3 | 0 | 1 | 0 |
| Papillae | 0 | 0 | 0 | 0 | 0 | 0 | 0 | 0 | 0 | 0 | 0 | 0 | 3 | 9 | 4 | 7 | 0 | 0 | 0 | 21 | 0 | 12 | 9 | 21 | 0 | 20 | 0 | 7 | 0 |
| Rondel | 0 | 0 | 1 | 0 | 0 | 1 | 0 | 1 | 0 | 0 | 0 | 0 | 27 | 37 | 27 | 51 | 0 | 0 | 0 | 38 | 0 | 37 | 9 | 38 | 5 | 28 | 1 | 36 | 2 |
| Saddle | 0 | 0 | 1 | 0 | 0 | 2 | 2 | 2 | 0 | 0 | 1 | 0 | 10 | 4 | 5 | 4 | 0 | 0 | 1 | 9 | 0 | 39 | 14 | 9 | 1 | 40 | 0 | 5 | 10 |
| Rectangular | 0 | 1 | 1 | 0 | 0 | 0 | 2 | 2 | 0 | 0 | 0 | 0 | 16 | 28 | 15 | 15 | 0 | 0 | 1 | 20 | 0 | 6 | 9 | 20 | 3 | 9 | 4 | 20 | 9 |
| Trapeziform | 0 | 0 | 0 | 0 | 0 | 0 | 0 | 0 | 0 | 0 | 0 | 0 | 6 | 11 | 10 | 20 | 0 | 0 | 0 | 7 | 0 | 4 | 0 | 7 | 0 | 1 | 0 | 3 | 2 |
| Trap. Polylobate | 0 | 0 | 0 | 0 | 0 | 1 | 0 | 1 | 0 | 0 | 0 | 0 | 15 | 8 | 9 | 8 | 0 | 0 | 0 | 0 | 0 | 12 | 1 | 0 | 0 | 8 | 0 | 4 | 0 |
| Trap. Sinuate | 0 | 0 | 0 | 0 | 0 | 1 | 0 | 1 | 0 | 0 | 1 | 0 | 26 | 10 | 10 | 9 | 0 | 0 | 0 | 9 | 0 | 13 | 7 | 9 | 0 | 17 | 0 | 7 | 1 |
| Hair base | 0 | 0 | 0 | 0 | 0 | 0 | 0 | 0 | 0 | 0 | 0 | 0 | 0 | 9 | 8 | 6 | 0 | 0 | 0 | 14 | 0 | 5 | 11 | 14 | 3 | 16 | 0 | 8 | 10 |
| Trichome | 0 | 0 | 0 | 0 | 0 | 0 | 0 | 0 | 0 | 0 | 1 | 0 | 5 | 7 | 8 | 9 | 0 | 0 | 1 | 9 | 0 | 10 | 9 | 9 | 5 | 2 | 0 | 5 | 8 |
| Stoma | 0 | 0 | 0 | 0 | 0 | 0 | 0 | 0 | 0 | 0 | 0 | 0 | 2 | 3 | 2 | 0 | 0 | 0 | 0 | 2 | 0 | 16 | 4 | 2 | 4 | 7 | 1 | 0 | 2 |
| Bulliform | 0 | 0 | 0 | 0 | 0 | 0 | 8 | 3 | 0 | 0 | 2 | 0 | 12 | 9 | 8 | 12 | 0 | 0 | 0 | 7 | 0 | 18 | 6 | 7 | 6 | 29 | 0 | 7 | 4 |
| Globular | 0 | 0 | 0 | 0 | 0 | 0 | 0 | 0 | 0 | 0 | 0 | 0 | 1 | 1 | 0 | 2 | 1 | 0 | 15 | 0 | 0 | 0 | 0 | 0 | 0 | 0 | 0 | 2 | 0 |
| Glob. Echinate | 0 | 0 | 0 | 0 | 0 | 0 | 0 | 0 | 0 | 0 | 0 | 0 | 0 | 0 | 1 | 0 | 0 | 0 | 0 | 0 | 0 | 1 | 0 | 0 | 0 | 0 | 0 | 0 | 0 |
| Scallop | 0 | 0 | 0 | 0 | 0 | 0 | 0 | 0 | 0 | 0 | 0 | 0 | 0 | 2 | 3 | 0 | 0 | 0 | 0 | 0 | 0 | 0 | 0 | 0 | 0 | 0 | 0 | 0 | 0 |
| Dicot. | 0 | 0 | 0 | 0 | 0 | 0 | 0 | 0 | 0 | 0 | 0 | 0 | 0 | 1 | 3 | 3 | 0 | 0 | 0 | 1 | 0 | 0 | 0 | 1 | 0 | 0 | 0 | 3 | 0 |
| Cyp. Papillae | 0 | 5 | 7 | 0 | 0 | 0 | 0 | 5 | 0 | 0 | 11 | 0 | 5 | 0 | 6 | 4 | 0 | 0 | 0 | 9 | 0 | 13 | 17 | 9 | 10 | 14 | 0 | 4 | 6 |
| Total | 0 | 5 | 7 | 0 | 0 | 8 | 51 | 22 | 0 | 0 | 38 | 0 | 250 | 250 | 250 | 250 | 1 | 0 | 21 | 9 | 0 | 246 | 148 | 250 | 105 | 240 | 17 | 250 | 99 |
| Silica Sk. Cells/ID | 0 | 0 | 0 | 0 | 0 | 1 | 3 | 2 | 0 | 0 | 0 | 0 | 17 | 21 | 12 | 11 | 1 | 0 | 0 | 17 | 0 | 13 | 2 | 17 | 3 | 5 | 1 | 15 | 10 |
| Echinates | 0 | 0 | 0 | 0 | 0 | 3 | 4 | 7 | 0 | 0 | 0 | 0 | 25 | 4 | 2 | 3 | 0 | 0 | 15 | 15 | 0 | 0 | 5 | 15 | 0 | 0 | 0 | 0 | 2 |
| Dendritics | 0 | 0 | 0 | 0 | 0 | 0 | 0 | 0 | 0 | 0 | 0 | 0 | 9 | 25 | 17 | 15 | 0 | 0 | 0 | 3 | 0 | 0 | 0 | 3 | 4 | 0 | 0 | 16 | 21 |
| Long Smooth | 0 | 0 | 0 | 0 | 0 | 0 | 0 | 0 | 0 | 0 | 0 | 0 | 16 | 0 | 3 | 0 | 0 | 0 | 0 | 3 | 0 | 0 | 0 | 3 | 0 | 0 | 0 | 5 | 0 |
| Long non-Smooth | 0 | 0 | 0 | 0 | 0 | 0 | 3 | 0 | 0 | 0 | 0 | 0 | 0 | 3 | 3 | 0 | 0 | 0 | 0 | 4 | 0 | 0 | 0 | 4 | 0 | 0 | 0 | 10 | 0 |
| Undulate/wavy | 0 | 0 | 0 | 0 | 0 | 3 | 0 | 3 | 0 | 0 | 0 | 0 | 15 | 21 | 0 | 6 | 5 | 0 | 22 | 22 | 0 | 21 | 2 | 22 | 9 | 9 | 2 | 5 | 0 |
| Undulate ll | 0 | 0 | 0 | 0 | 0 | 0 | 0 | 3 | 0 | 0 | 0 | 0 | 11 | 13 | 9 | 3 | 0 | 0 | 0 | 18 | 0 | 20 | 0 | 18 | 7 | 7 | 0 | 3 | 8 |
| Undulate η | 0 | 0 | 0 | 0 | 0 | 1 | 0 | 1 | 0 | 0 | 0 | 0 | 11 | 5 | 0 | 0 | 0 | 0 | 0 | 0 | 0 | 20 | 0 | 0 | 0 | 0 | 0 | 7 | 0 |
| Papillae | 0 | 0 | 0 | 0 | 0 | 0 | 0 | 2 | 0 | 0 | 0 | 0 | 10 | 5 | 0 | 0 | 0 | 0 | 0 | 0 | 0 | 0 | 0 | 0 | 0 | 0 | 0 | 4 | 5 |
| Rondel | 0 | 0 | 0 | 0 | 0 | 1 | 0 | 1 | 0 | 0 | 0 | 0 | 5 | 8 | 1 | 8 | 1 | 0 | 9 | 9 | 0 | 7 | 0 | 9 | 0 | 1 | 0 | 2 | 0 |
| Hair base | 0 | 0 | 0 | 0 | 0 | 2 | 0 | 0 | 0 | 0 | 0 | 0 | 1 | 0 | 0 | 0 | 0 | 0 | 0 | 9 | 0 | 0 | 0 | 9 | 0 | 0 | 0 | 2 | 1 |
| Trichome | 0 | 0 | 0 | 0 | 0 | 0 | 0 | 0 | 0 | 0 | 0 | 0 | 13 | 8 | 4 | 8 | 0 | 0 | 0 | 0 | 0 | 7 | 0 | 0 | 0 | 0 | 0 | 6 | 4 |
| Stoma | 0 | 0 | 0 | 0 | 0 | 0 | 0 | 0 | 0 | 0 | 0 | 0 | 3 | 0 | 0 | 0 | 0 | 0 | 0 | 0 | 0 | 7 | 0 | 0 | 3 | 0 | 0 | 0 | 0 |
| Cyperaceae | 0 | 0 | 0 | 0 | 0 | 0 | 0 | 0 | 0 | 0 | 0 | 0 | 16 | 8 | 3 | 8 | 0 | 0 | 0 | 4 | 0 | 0 | 0 | 4 | 0 | 0 | 0 | 0 | 0 |
| Dicot | 0 | 0 | 0 | 0 | 0 | 0 | 0 | 0 | 0 | 0 | 0 | 0 | 3 | 0 | 3 | 0 | 0 | 0 | 0 | 0 | 0 | 0 | 0 | 0 | 0 | 0 | 0 | 3 | 0 |
| Indeterminate | 0 | 0 | 0 | 0 | 0 | 0 | 3 | 0 | 0 | 0 | 0 | 0 | 16 | 3 | 3 | 8 | 0 | 0 | 0 | 4 | 0 | 0 | 0 | 4 | 0 | 0 | 0 | 0 | 0 |
| Total cells | 0 | 0 | 0 | 0 | 0 | 3 | 7 | 15 | 0 | 0 | 0 | 0 | 108 | 87 | 42 | 35 | 0 | 0 | 0 | 74 | 0 | 48 | 0 | 74 | 11 | 17 | 2 | 61 | 41 |
| Dic. Leaf | 0 | 0 | 0 | 0 | 0 | 0 | 0 | 0 | 0 | 0 | 0 | 0 | 0 | 0 | 0 | 0 | 0 | 0 | 0 | 1 | 0 | 0 | 0 | 1 | 0 | 0 | 0 | 0 | 0 |
| Poaceae Culm/Leaf | 0 | 0 | 0 | 0 | 0 | 0 | 0 | 0 | 0 | 0 | 0 | 0 | 4 | 4 | 2 | 2 | 1 | 0 | 0 | 0 | 0 | 1 | 2 | 0 | 0 | 9 | 0 | 4 | 1 |
| Triticeae Inflo. | 0 | 0 | 0 | 0 | 0 | 1 | 1 | 1 | 0 | 0 | 0 | 0 | 4 | 5 | 2 | 4 | 0 | 0 | 0 | 0 | 0 | 0 | 0 | 0 | 4 | 7 | 0 | 3 | 2 |
| Phragmites Culms | 0 | 0 | 0 | 0 | 0 | 0 | 0 | 0 | 0 | 0 | 0 | 0 | 0 | 1 | 0 | 0 | 0 | 0 | 0 | 0 | 0 | 0 | 0 | 0 | 0 | 0 | 1 | 0 | 0 |
| Wild grass Inflo. | 0 | 0 | 0 | 0 | 0 | 0 | 0 | 0 | 0 | 0 | 0 | 0 | 3 | 2 | 2 | 1 | 0 | 0 | 0 | 5 | 0 | 1 | 1 | 5 | 0 | 1 | 0 | 0 | 0 |
| Panicoid type | 0 | 0 | 0 | 0 | 0 | 0 | 0 | 0 | 0 | 0 | 0 | 0 | 4 | 2 | 0 | 0 | 0 | 0 | 0 | 2 | 0 | 6 | 0 | 2 | 0 | 1 | 0 | 2 | 1 |
| Indeterminate | 0 | 0 | 0 | 0 | 0 | 1 | 2 | 1 | 0 | 0 | 0 | 0 | 4 | 7 | 3 | 4 | 1 | 0 | 0 | 6 | 0 | 5 | 1 | 6 | 2 | 4 | 0 | 6 | 0 |

Recorded single cells, silica skeletons, and plant anatomical parts from each sample.

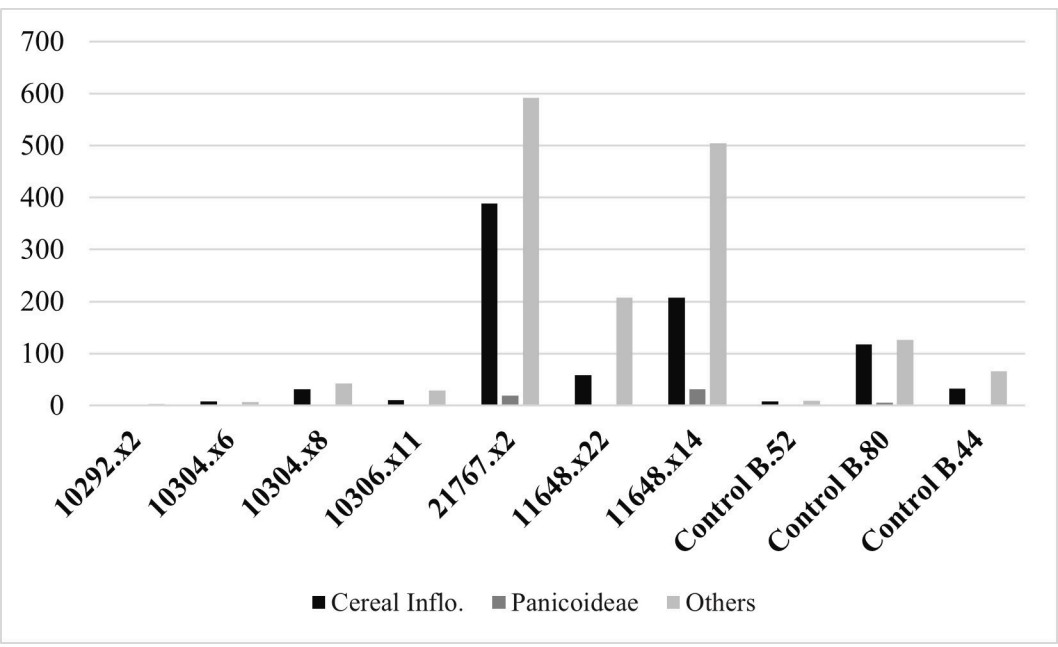

**Fig 8. Phytoliths groups by artefact.** Raw count comparison of phytoliths from cereals, Panicoideae and others, including cells from leaves and culms.

Morphometry was conducted on dendritics from the quern in Building 80 and one grinder (11648.x14) from Building 44. Based on our reference collection and published data, the dendritics can be attributed to *Triticeae* falling in the range of *T. dicoccum* (emmer) and *T. monococcum*

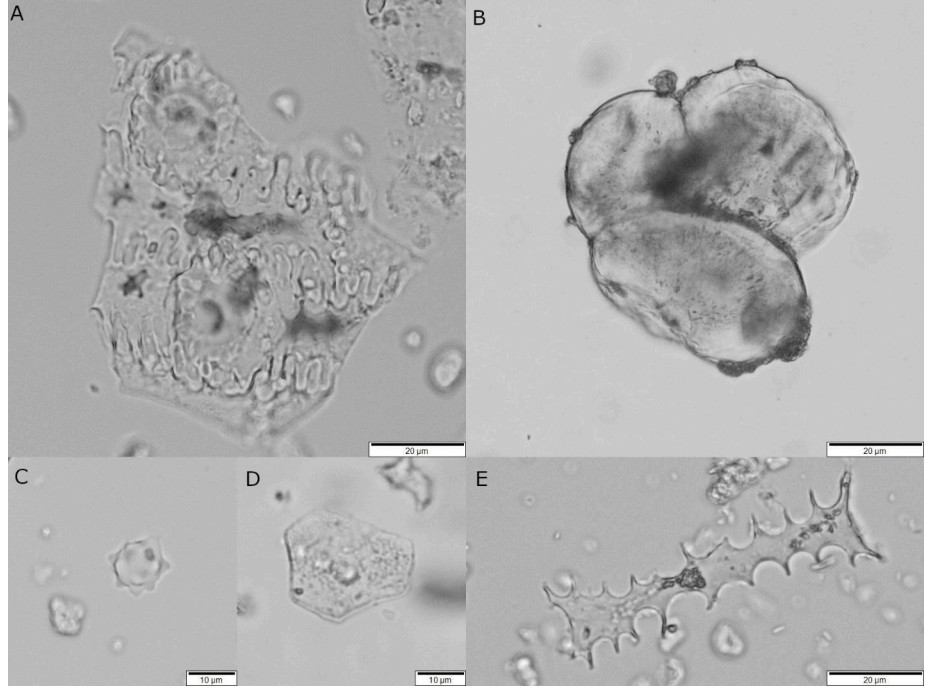

**Fig 9. Phytoliths recovered in this study.** A) silica skeleton from Panocoideae inflorescence; B) globular psilate mesophyll cells; C) globular echinate; D) papillae cell "cone" from Cyperaceae; E) Long dendritic cell from Triticeae. Pictures at 40x magnification.

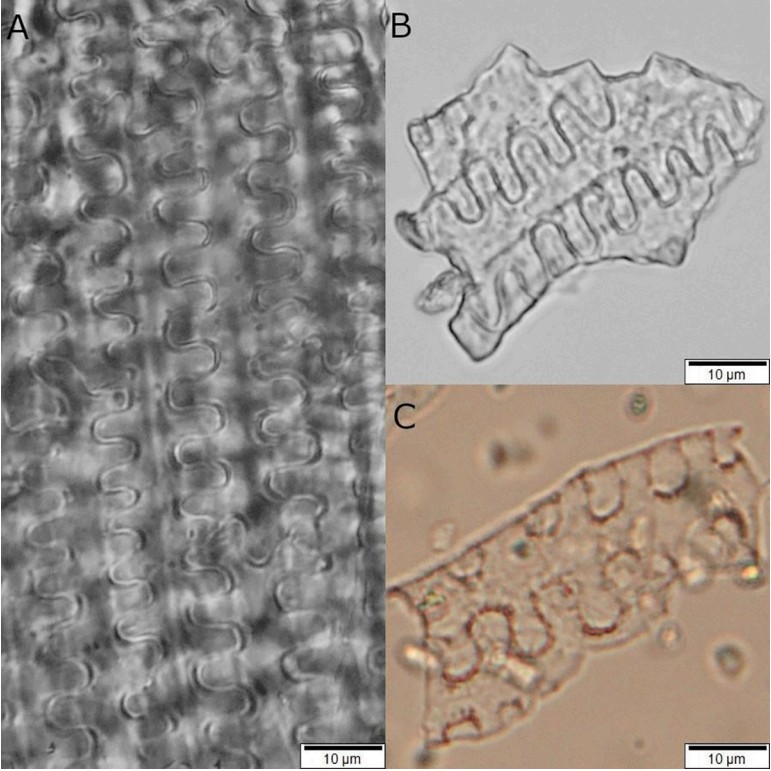

**Fig 10. Phytolith examples of long cells with ∩ ornamentation.** A) Modern upper lema *S. verticillata*; B-C) Archaeological examples recovered in this study. Pictures at 40x magnification.

(einkorn) [19, 53]. A total of 134 silica skeletons from anatomical parts such as inflorescence, culms and leaves of grasses were observed. Most of them were highly fragmented and affected by taphonomy. In some cases, silica skeletons presented straight transversal cuts (Fig 10B and 10C), probably caused by post-harvesting processes such as trampling and dehusking [175, 176].

Long (elongate lobated) and short (crosses) cell phytoliths from *Panicoideae* were also observed together with silica skeletons with long cells with ∩ and η ornamentations. These types of ornamentation are normally found in *Panicoideae* and other C4 grasses inflorescence [47, 177–180] and more specifically *Setaria verticillata* (Fig 10). For this reason, we made morphometric comparisons (based on characters from Zhang et al. [125] between ∩ patterns from modern *Setaria verticillata* from our reference collection and the recovered archaeological material showing similar patterns. For extra safety, we performed the same measures on phytoliths from *Phragmites* culms since previous studies noticed that this plant is widely present at the site [79, 181] and that phytoliths with similar ornamentation can occur on reed culms as well. The results of our analysis show that the silica skeletons from Çatalhöyük are compatible with those of modern *S. verticillata*. (Table 4).

## 4.0 Discussion

### 4.1 Hidden plants and hidden cycles–some aspects of underground storage organs' exploitation and seasonality

The combined analysis of starch grains and phytoliths extracted directly from the artefacts has allowed us to broaden the spectrum of plants present at the site. In particular, starch grain analysis has strengthened our understanding of the possible use of underground organs such

**Table 4. Morphometric results.**

| | Dendritic long cells Largest width from samples and modern reference | | | |
| --- | --- | --- | --- | --- |
| | Range | Mean | Standard Deviation | |
| **21767.x2 s15** | 14.90–27.97 μm | 19.2458 μm | 4.541 μm | N = 12 |
| **S.42** | 15.46–37.18 μm | 25.2955 μm | 5.5898 μm | N = 20 |
| ***T. aestivum*** | 12.93–36.96 μm | 23.2052 μm | 5.8323 μm | N = 50 |
| ***T. dicoccum*** | 17.92–35.82 μm | 26.7944 μm | 4.5392 μm | N = 50 |
| ***T. monococcum*** | 9.36–21.96 μm | 17.2534 μm | 2.4463 μm | N = 50 |
| ***H. vulgare*** | 7.03–22.27 μm | 13.5296 μm | 3.2981 μm | N = 50 |
| | Long Undulate ∩ Cells | | | |
| ***Phragmites*** | H = 3.90 ± .83μm, W = 13.56 ± 1.96 μm, R = (0.29 ± 0.42) | | | N = 100 |
| ***S. verticillata*** | H = 6.25 ± 1.55 μm, W = 16.49 ± 3.34 μm, R = (0.38 ± 0.46) | | | N = 100 |
| ***Çatalhöyük*** | H = 5.91 ± 1.75 μm, W = 15.11 ± 3.67 μm, R = (0.39 ± 0.47) | | | N = 26 |

N refers to the number of measurements, R refers to the ratio between H/W.

Comparison between modern references and archaeological samples.

as bulbs and rhizomes of plants from the *Iridaceae* and *Liliaceae* families. These geophytes are quite common in the flora of Turkey, represented by 10 genera [182]. Some of these species are traditionally appreciated as a food source in the region and it is therefore interesting to observe their presence on plant processing artefacts from Çatalhöyük. Starch with general characteristics similar to the one originating from *Cyperaceae* (sedges) plants has also been observed in the sampled implements. Carbonized rhizomes/tubers of club-rush (*Bolboschoenus glaucus* (Lam.) S.G.Sm.) have been identified in the macroremains assemblage at Çatalhöyük, and they represent the third most common plant group after cereals and pulses at the site [80]. *Typha* rhizomes have not been observed in the macroremains, although there are various archaeological precedents for its use and processing in prehistoric times [146–148]. Even though it was not possible to refine their identification at species level, the richness and diversity of starch grains originating from underground storage organs is noticeable at Çatalhöyük. The above clearly indicate that a wide range of geophytes species were processed with lithic tools traditionally associated with the grinding of cereals and seeds. Some of these taxa have restricted seasonal cycles, and ethnographic accounts from Turkey and other areas record the use of *Iridaceae* and *Liliaceae* species in the spring, when the plants are blooming and are easily identifiable. Many of these species and other geophytes are consumed raw, ground into flour or cooked [29, 183–191] (Table 5). The role of such resources could have been substantial for the caloric intake considering that by spring, the food stored from the previous year's harvest, such as cereals and pulses, might have been running low. Then, the incorporation of raw or processed USOs into the diet would have balanced the shortage of carbohydrates from crops. Moreover, ethnobotanical accounts mention that the majority of the fresh corms and bulbs (e.g., *Crocus ssp*., *Iris ssp*. and *Tulipa ssp*.) are consumed while out in the field and they rarely enter the domestic space [185, 188, 192]. However, the presence of starch from these plant groups on the grinding tools at Çatalhöyük suggests that at least part of these resources were processed in the houses as part of a very extended use of the available plant resources. It is worth noting that some *Liliaceae* and *Iridaceae* species contain toxins and need to be processed for detoxification before consumption, or they are used in small quantities for medicinal purposes. This is the case for species of the genus such as *Fritillaria*, *Lilium* and *Tulipa*, whose USOs can be crushed and powdered and used in folk medicine in parts of current Turkey for treating wounds and other ailments [190, 193]. Although we cannot make a direct inference

**Table 5. Some economically important geophytes in Turkey and the surrounding regions.**

| Taxon | Parts | Uses | Location | References |
|---|---|---|---|---|
| **Iridaceae** | | | | |
| *Crocus* | | | | |
| *C. ancyrensis* (Herb.) Maw. | Flower/ USO | Food | Turkey | [185, 210, 211]. |
| *C. biflorus* Miller subsp. *tauri* Mathew. | USO | Food | Turkey/Iraq | [188, 210, 212]. |
| *C. cancellatus* Herbert subsp. *Damacenus.* | USO | Food | Turkey | [189, 196, 213]. |
| *C. danfordiae* Maw | USO | Medicine | Turkey | [214]. |
| *C. graveolens* Boiss. & Reut. | Aerial parts | Medicine | Turkey | [210, 213]. |
| *C. kotschyanus* subsp. *Kotschyanus* K.Koch. | Aerial parts | Medicine | Turkey | [210, 215]. |
| *C. olivieri* J.Gay | USO | Medicine | Turkey | [216]. |
| *C. pallasii* Goldb. | USO | Food | Turkey | [196, 213]. |
| *C. pallasii* Goldb. subsp. *Turcicus* B. Mathew. | USO | Food | Turkey | [196]. |
| *C. reticulatus* Steven ex Adam. | Flower | Food | Caucasus | [211]. |
| *C. sativus* L. | Flower/ USO | Food/Dye | Turkey | [210, 216, 217]. |
| *C. vitellinus* Wahlenb. | Aerial parts | Medicine | Turkey | [210]. |
| *Crocus* ssp. | USO | Food | Armenia | [188]. |
| *Gladiolus* | | | | |
| *G. atroviolaceus* Boiss. | USO | Food/ Medicine | Turkey | [184, 192, 210, 213, 218]. |
| *G. illyricus* W. Koch. | USO | Food | Turkey | [210, 212]. |
| *G. italicus* Mill. | Flower | Medicine | Turkey | [210]. |
| *G. kotschyanus* Boiss. | USO | Food/ Medicine | Turkey | [184, 213]. |
| *Iris* | | | | |
| *I. barnumiae* Foster & Baker. | Leaf | Food | Turkey | [219]. |
| *I. caucasica* Hoffm. | Tepals | Food/ Medicine | Turkey | [184, 195, 210, 220]. |
| *I. galatica* Siehe. | Whole | Food | Turkey | [185, 192, 221]. |
| *I. germanica* L. | USO | Food | Turkey | [184, 210]. |
| *I. iberica* Hoffm. subsp. *Elegantissima* (Sosn.) Fed. Takht. | Tepals | Food | Turkey | [184, 210]. |
| *I. masiae* Leichtlin ex Dykes. | Flower | Food | Turkey | [213]. |
| *I. paradoxa* Steven. | USO | Medicine | Turkey | [210]. |
| *I. persica* L. | Tepals | Food | Turkey | [189, 196, 213]. |
| *I. reticulata* M. Bieb. | Tepals | Food/ Medicine | Turkey | [189, 196, 210, 213, 218]. |
| *I. sari* Schott ex Baker. | Whole | Medicine | Turkey | [210, 220]. |
| *Romulea* | | | | |
| *R. tempskyana* Freyn. | USO | Food | Turkey | [222]. |
| **Liliaceae** | | | | |
| *Fritillaria* | | | | |
| *F. collina* Adams. | Flower | Food | Caucasus | [211]. |
| *F. crassifolia* Boiss. & Huet. subsp. *kurdica* (Boiss. & Noe) | USO | Medicine | Turkey | [223]. |
| *F. pinardii* Boiss. | USO | Food/ Medicine | Turkey | [195]. |

(*Continued*)

**Table 5.** (Continued)

| Taxon | Parts | Uses | Location | References |
|---|---|---|---|---|
| *F. wendelboi* (Rix) Teksen | Whole | Medicine | Turkey | [190]. |
| ***Gagea*** | | | | |
| *G. granatellii* (Parl.) Parl. | Whole | Fodder | Turkey | [185]. |
| *G. villosa* (M.Bieb.). | USO | Food | Turkey | [189]. |
| ***Lilium*** | | | | |
| *L. candidum* L. | Flower/USO | Medicine | Israel | [224]. |
| *L. martagon* L. | USO | Food/Medicine | Turkey | [194]. |
| ***Tulipa*** | | | | |
| *T. armena* Boiss. | USO | Food | Turkey | [188, 218, 219]. |
| *T. armena* var. *lycica* (Baker) Marais. | Flower/USO | Food/Medicine | Turkey | [185, 195]. |
| *T. cinnabarina* K.Perss. | USO | Medicine | Turkey | [190]. |
| *T. humilis* Herb. | Flower | Others | Turkey | [185]. |
| *T. julia* K.Koch. | Whole | Others | Turkey | [195, 213]. |
| *T. montana* Lindl | USO | Food | Iraq | [188]. |
| *T. orphanidea* Boiss. ex Heldr. | USO | Food | Turkey | [187]. |
| **Cyperaceae** | | | | |
| ***Cyperus*** | | | | |
| *C. longus* L. | USO | Medicine | Turkey | [213]. |
| *C. rotundus* L. | USO | Food/Medicine | Turkey | [213, 214, 220, 225, 226]. |
| ***Carex*** | | | | |
| *C. divulsa* Stokes ssp. *divulsa*. | Leaf | Crafts | Turkey | [185]. |
| **Typhaceae** | | | | |
| ***Typha*** | | | | |
| *T. angustifolia*. | Whole | Crafts | Turkey/Balkans | [200, 213]. |
| *T. laxmannii* Lepechin. | Whole | Food/Crafts | Turkey | [185, 201]. |

List of Iridaceae and Liliaceae species, and other geophytes mentioned in this study, along with their use and part of interest. Other refers to multiple uses such as fodder and ornamental.

for similar uses of such plants in the Neolithic, the exploitation of such geophytes by the Çatalhöyük community seems to reflect a rich phytocultural knowledge and possibly the development of complex culinary practices [29, 184, 194–196].

Other geophytes mentioned in this study such as *Typhaceae* and *Cyperaceae* are known to produce starchy flour [5, 143, 146]. Experiments on processing club-rush tubers have demonstrated that pulverization by grinding is a crucial step for boosting their full nutritional value [197]. The flour can be later eaten as gruel or bread, the latter mixed with cereal flour to get the consistency needed for baking [5, 198]. This practice seems to be present at Çatalhöyük where fragments of *Cyperaceae* tubers were identified embedded in carbonized food remains [84]. The processing of *Cyperaceae* tubers can be laborious, time-consuming and requires equipment and processing methods similar to those used for cereal processing [5, 197, 199], therefore possibly explaining the presence of *Cyperaceae* starch on these artefacts. This presence also suggesting that the final stage of processing of such plants was carried out on-site.

Apart from being used as food source, *Cyperaceae* and *Typhaceae* fibers and leaves can be used for making strings, matting and other crafts [143, 185, 200]. There are reports in Turkey for the collection of these plants for matting and crafting in late autumn and winter [192, 201–203]. The same temporality can be suggested for Çatalhöyük due to the on-site presence of juvenile micro-mollusks, which may have arrived at the households along with water plant roots, and that show a growing stage that occurs between the end of summer and mid-autumn, [74, 204]. This season also coincides with the time of the year when carbohydrates in *Typha* and *Cyperaceae* USOs are at the highest levels [205, 206], making this the most convenient period for the collection of these plants for both crafting and food. Therefore, considering the complex preparations of foodstuffs and crafts that arise from our analyses, geophyte taxa may have played a greater role throughout the year either as food ingredients or craft raw materials at the site than previously considered. This new information highlights the extensive knowledge the people of Çatalhöyük had on wild plant and their seasonal availability. This broad-spectrum approach in respect to the use of plants is further supported by the evidence from the carbonized material and suggests a wide agro-ecological understanding of the Çatalhöyük community of their immediate landscape [76, 81, 207] as well as of more distant ecosystems [83, 203, 208, 209].

## 4.2 Small millets on the menu? Some clues, problems and interpretations

Another interesting find in the Çatalhöyük microbotanical remains assemblage is the presence of *Panicoideae* starch grains that we suggest might have originated from a type of local small millet. Although small millets are usually seen nowadays as famine food [227], they are consumed and appreciated in various regions of the world [45, 228]. The morphometric analysis of long cell phytoliths with Ո ornamentation along with the starch data and the secured provenance of the microremains analyzed on this study suggests the presence of *S. verticillata* at Çatalhöyük. Nowadays, bristly foxtail is consumed in some regions of Africa and India where the seeds are ground for making bread, porridge and beer [227, 229, 230].

Despite the substantial archaeobotanical evidence for the use of wild small millet species, such as *Setaria* and *Panicum* genera, at other prehistoric sites in Anatolia and surrounding regions (i.e., [46, 231–233]), bristlegrass and other small millets are absent in the macrobotanical assemblage at Çatalhöyük (i.e. [76, 83, 85]). One possible cause could be a preservation bias since small millets are vulnerable to charring conditions making them less likely to be preserved by this means than other cereals [46, 234–238]. It is also possible that small millets, being a seasonal resource used when available in the landscape, were treated differently to many other wild resources which were instead stored, thus making them less susceptible to preservation by accidental charring (i.e. [237, 239, 240]).

Despite the absence of macrobotanical remains at the site, it is clear from the microbotanical data that Panicoid grasses played an important role, both as raw material used in basketry [202] and, with all probability, as supplementary (seasonal) food. A clue for this can be found in the isotope analyses. Signals of $C_4$ plants have been found in herbivores [241, 242], probably originating from the grazing of sedges and other small-seeded grasses. Indeed, seeds of $C_4$ plants have been found in sheep/goat dung pellets [80]. Considering that ovicaprid were among the most common animal protein available for the human population [242], it can be assumed that they might have contributed to the $C_4$ signal in humans. However, recent studies have shown that apparently this was not the case and that the consumption of $C_4$-eating animals causes little impact on human carbon isotope values [242]. Despite that the most recent isotope analyses on human remains have not been conclusive [242], previous studies have brought to attention possible incorporation of unrecognized $C_4$ plants into some individuals'

diets, and this has been interpreted to mean that these people were eating a different diet and they were more prone to exploiting seasonal resources [241, 243, 244]. In terms of procurement, small millets tend to have asynchronous ripening time which extend the availability of these plants until the end of autumn [230], even after the harvesting period of wheat and barley in the summer and autumn months [80, 85, 192, 203]. Thus, *S. verticillata* and small millets in general can function as a reliable food source (along with USOs), as buffering during periods of shortage due to, for instance, harvest failure [207], or as supplementary food that can be collected (along with USOs) even when carrying out other tasks in the fields or the surrounding countryside (i.e. [188]).

## 4.3 Some inferences about on-site post-harvesting processes

Plant procurement strategies at Çatalhöyük clearly expanded the boundaries of farmed resources and included the use of a wide array of wild plants as the microremains demonstrate. Furthermore, starch and phytoliths from grinding tools make apparent that such implements played a key role in transforming both domestic and wild plant resources into food by improving their digestion, nutritional value, or detoxifying them. The processing of plant foods formed part of the daily routines, as evidenced by stress markers on human bones at the site [245]. Supporting this, we observed in the microbotanical data some patterns that could be linked to the tool's functionality and for instance, to on-site post-harvesting practices such as dehusking, grinding and possible clues for the use of plant-based crafts and implements.

**4.3.1 Dehusking.**   The quern found in Building 80 produced the highest phytolith yield with a predominance of cells and silica skeletons from *Triticeae* inflorescence, similar to patterns observed in experimental tools used for dehusking [10]. The very low presence of starch grains on the tool (n = 11) and the complete absence of starch from the surrounding sediments (see [246]), together with the phytolith evidence, strongly support that this implement was used *in situ* for dehusking or beating spikelets for separating seeds from the chaff in cereals (see [10, 247, 248]). The use of the quern for plant processing activities is further corroborated by the results of the microwear analysis. The use-face shows smoothing but no intense leveling of the topography is otherwise observed. Microscopic observations at high magnifications (100x, 200x) include a fine, granular, reflective micropolish occasionally intermingled with localized spots of smooth micropolish; it develops on the higher elevations of the microtopography, and is distributed across the use-face with a longitudinal directionality (Fig 11E and Table 6). This type of micropolish is consistent with contact with non-greasy plant material (e.g., [35, 45, 249]), but due to the rejuvenation of the use-face and the overall condition of the tool surface a more precise interpretation of the type of plant material was not possible.

**4.3.2 Plant processing and other processes.**   In the case of Building 52, phytoliths are almost absent on the upper tools (10292.x2 & 10304.x6), while the lower tools have higher concentrations of phytoliths from *Triticeae* inflorescences. However, starch recovery from upper and lower grinding tools was similar in terms of richness and diversity of types including cereals, legumes and USOs, suggesting that these tools were used for diverse processing activities such as grinding, light pounding and mashing of either clean or partially cleaned cereals and wild plants. Moreover, moprhometrical and technological analyses of the grinding tools along with the observed microbotanical variability between upper and lower tools clearly suggest that these tools did not form part of the same toolkit, and instead they were used separately for different stages of plant processing. Contrary to the grinders, the lower slabs were also used for grinding partially cleaned cereals or dehusking, explaining the differences in inflorescence phytoliths among these two artefact categories.

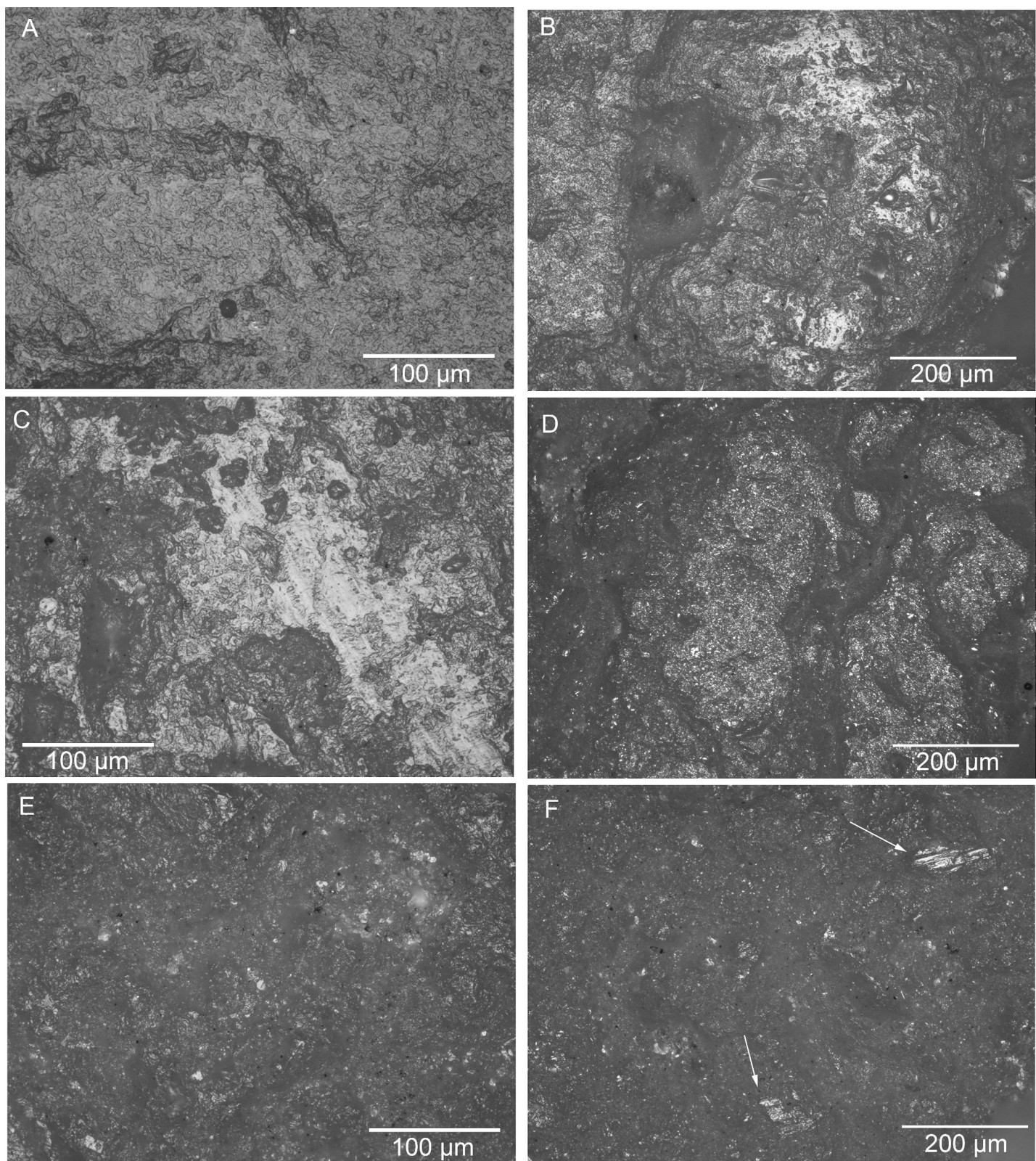

**Fig 11. Microwear traces on grinding tools.** A) 11648.x22 clay/plaster micropolish (200x); B) 11648.x22 woody plant micropolish (100x); C) 11648.x14 mineral micropolish (200x); D) 11648.x8 plant (cereal) micropolish forming distinct patches (100x); E) 21767.x2 plant micropolish (200x); F) 10292.x2 plant (legumes) micropolish.

**Table 6. Grinding tools' microwear traces.**

| Object No. | Macrowear traces. | Microwear traces. | Contact Material. | Inorganic residues. |
|---|---|---|---|---|
| 10292. x2 | Levelling of grains. | Granular, reflective micropolish, H/L microtop; striated micropolish. | Plant (legumes) & stone on stone contact. | N/A |
| 10304. x6 | Levelling of surface topography, grain extraction. | Microfractures, sparse microstriations; (1) reflective smooth, pitted micropolish, sinuous/flat topography, H microtop; (2) granular, reflective, rough micropolish, smooth spots. | (1) Mineral (soft-medium hardness); (2) Plant (cereal). | N/A |
| 10304. x8 | Not sampled for microwear analysis. | Not sampled for microwear analysis. | N/A | No |
| 10306. x11 | Levelling of grains, grain extraction. | Not sampled for high-power analysis. | N/A | N/A |
| 21767. x2 | Levelling of grains. | Granular, reflective micropolish, smooth spots, not well developed, H microtop. | Non-greasy plant. | N/A |
| 11648. x8 | Intense levelling. | Granular, reflective micropolish, smooth spots, distinct patches, H microtop; striated micropolish. | Plant (cereal) & stone on stone contact. | N/A |
| 11648. x22 | Flat use-face: intense levelling, grain extraction; Convex use-face: smoothing of surface topography. | Flat use-face: (1) dull, rough micropolish, H/L microtop & intermediate area, microstriations; (2) smooth, pitted micropolish, sinuous topography, distribution on individual grains/aggregations of grains; (3) granular reflective micropolish. | (1) Clay/plaster; (2) Woody plant; (3) Plant. | Plaster trapped in pits. |
| 11648. x14 | Flat use-face: intense levelling, grain extraction. | Flat use face: (1) dull, rough micropolish, H/L microtop & intermediate area, microstriations; (2) smooth, pitted micropolish, sinuous topography, distribution on individual grains/aggregations of grains; (3) granular reflective micropolish; (4) patches of flat micropolish, microstriations. | (1) Clay/plaster; (2) Woody plant; (3) Plant; (4) Mineral. | Red-colour staining, possibly mineral. |

Summary of observed microwear traces (H/L microtop = higher & lower microtopography; H microtop = higher microtopography).

Further insights into the function of the upper grinding tools are offered by microwear analysis. Grinder 10292.x2 exhibits microwear traces consistent with contact with plant material (i.e., fine, granular, reflective micropolish that develops across the surface without forming distinctive patches and with a directionality perpendicular/slightly diagonal to the long axis of the tool) (Fig 11F and Table 6). These traces resemble traces obtained experimentally by grinding legumes (Fig 12). A striated micropolish observed on grinder 10292.x2 resulted from stone on stone contact between the upper and the lower grinding tool during the grinding process. Mixed wear signatures on the flat use-face of tool 10304.x6 suggest contact with mineral material of soft-medium hardness as well as contact with a material of plant origin. This tool has multiple use-faces and may have been used as a multi-functional tool for plant and non-plant related activities. Observed wear traces include intense levelling of the surface topography accompanied by grain extraction. At high magnifications, wear patterns include microfractures on grain crystals and a reflective micropolish, of flat/slightly sinuous topography, smooth

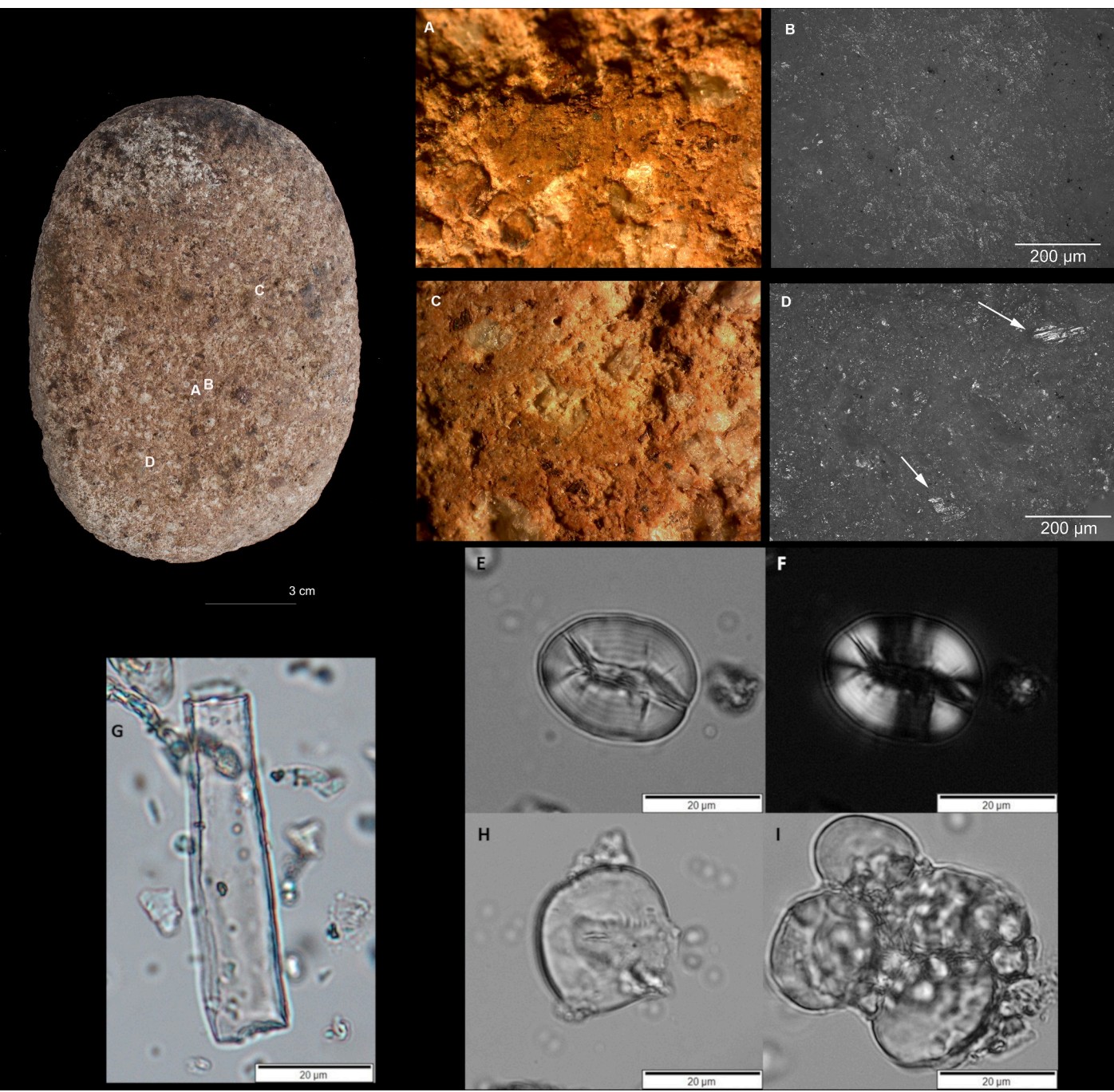

**Fig 12. Microwear traces and microbotanical residues on the use-face of grinder 10292.x2.** A) levelling of grains (magnification 30x); B) plant (legumes) micropolish; C) low power observations levelling of grains (magnification 20x); D) plant (legumes) micropolish and; G) Elongate smooth psilate phytolith; H) Triticeae starch grain damaged by pressure; I) Triticeae starches aggregate. Microbotanical pictures at 400x magnification.

texture and pitted appearance that develops in patches of random disposition. Microstriations accompany the micropolish. The mineral micropolish identified on this tool does not resemble the striated micropolish that is associated with the stone on stone contact usually seen on grinding tools. In addition, a granular, reflective and rough micropolish that has a reticular distribution and forms distinctive patches is present on the tool use-face. In places, localized

spots with smooth-textured micropolish are present within the patches. These wear traces have certain similarities with experimentally-produced traces that result from cereal grinding [35, 133].

In Building 44, grinder 11648.x8 has no phytoliths and the only starch observed was from legumes and USOs, along with microwear traces consistent with plant processing (Fig 11D and Table 6). While the microbotanical remains seem to suggest an exclusive use for processing non-cereal resources, the microwear traces are consistent with cereal grinding. Apart from the similar phytolith assemblage on the remaining two artefacts from this building, grinder 11648.x22 stands out due to the high presence of globular/spheroid psilate phytoliths of the type produced in leaves mesophyll, suggesting the use of this artefact for processing woody plants leaves whether for crafts or mashing for obtaining secondary products such as oils [215, 250]. While wear patterns observed on 11648.x22 (Fig 11A, 11B and Table 6) and on 11648.x14 (Fig 11C and Table 6) support the use of these tools for plant-processing activities including the processing of woody plant materials, both tools also show a well-developed, dull micropolish of rough texture and irregular topography that affects the higher microtopography, intermediate area, and in places also penetrates the lower microtopography. It develops across the surface in interconnected patches (medium/high density) and is accompanied by microstriations. This type of micro-polish presents similarities with microwear traces observed on experimental tools used for burnishing leatherhard clay [89] (Fig 3) and on archaeological plastering tools [251]. At Çatalhöyük, soft-lime and mainly white marl—a highly calcareous clay locally known as *Ak Toprak*—was used to plaster the house surfaces [252–254].The use of these tools for different activities resulted in mixed wear signatures that hinder more detailed characterization of the micropolish associated with plant contact. In the case of the woody plant materials, based on the characteristics of the use-face of the tools (intense levelling and smoothing that extends across the whole surface of the tool), their presence on the tool surfaces is more consistent with grinding activities. The presence of wood and wood-like materials has also been noted in other studies of grinding tools [35, 137]. Hamon and colleagues [137] have suggested that the grinding of wood and bark materials may be associated with the acquisition of powdered materials with colouring or medicinal properties, or potentially fruit processing. Continuing microwear analysis of the Çatalhöyük grinding tools may shed more light on the type of processing associated with this type of contact material. Overall, the mixed wear signatures *in tandem* with the presence of plaster residues in the pits of 11648.x22, demonstrate the complex life histories of these objects.

**4.3.3 Clues to related plant-based crafts.** Phytoliths recovered from grinder 11648.x22 and the quern from Building 80 also included globular/spheroid echinates originating from palms (Fig 9). In the Mediterranean and Middle East regions, the occurrence of palm phytoliths in archaeological ground stone implements is not rare [10], and there are many pathways though which these phytoliths could have been incorporated into the artefacts. This can occur at different stages, before plant processing through the storage of cereals and artefacts in basketry containers made of palm leaves [255] or by coming into contact with other craft materials made of palm leaves such as mats that can be placed below the slabs as spill caches, baskets for transporting the grains, or utensils used during winnowing [1]. However, palm phytoliths a rare find probably because palms are not native to central Anatolia and the surrounding landscape of Çatalhöyük provided other and abundant materials for crafting baskets and mats, such as *Phragmites* and cattail [192, 201]. Rosen [92] also noticed that this 'exotic' type was associated with food-related and storage contexts, suggesting that it may have originated from *Phoenix dactylifera* leaves arriving in the form of basketry from the Levant through long-distance exchange networks. However, *Phoenix theophrasti*, a palm local to the Aegean coast and known in antiquity for its economic importance could also produce this morphotype [256,

257]. Wherever it was the geographical origin of palm, it is consistent for this good to arrive at the site via exchange networks with the Mediterranean world as also evidenced by the presence of sea mollusks [258].

## 5.0 Conclusion

The current work shows the importance of starch and phytolith analyses for understanding plant-processing activities and for identifying resources that are absent or underrepresented in the archaeobotanical macroremains record. Such an understanding is further enhanced by the integration of the results of the microbotanical analysis with the microwear study of grinding tools from where the remains were recovered. Despite that the tools examined represent only a small portion of the whole grinding tools assemblage at the site, our work made evident the wide spectrum of plant resources, both domesticated and wild, that the people of Çatalhöyük were using. The wild plant resources were collected from the different ecological niches present in the surrounding landscape and the gathering of such resources followed a seasonal calendar that took into consideration the labor bottlenecks from the cultivated fields (see also [74]). These resources were as important as the domesticated ones; they were used as a complement the main diet but they were also utilized as condiments, medicines and for producing crafts. Therefore, the results from our work complements and reinforces previous interpretations about the site's farming economy that was based on a mixed and flexible regime combining farmed products with locally available wild resources. However, the analysis of microremains significantly expanded the archaeobotanical record from Neolithic Çatalhöyük by revealing, in the food processing chain, the presence of geophyte species and panicoid grasses. More importantly, by employing multiple lines of evidence and by integrating microbotanical and microwear data with technological and contextual information, the function of different forms of grinding tools can be assessed in a more accurate way, allowing for a more nuanced understanding of plant use and processing at Neolithic Çatalhöyük.

## Supporting information

**S1 File.**
(XLSX)

## Acknowledgments

We would like to acknowledge the valuable contributions by previous microbotanical teams at Çatalhöyük whose investigations on phytoliths and starches at the site have provided a solid foundation for this study. We would like to thank all the team members of the Çatalhöyük Research Project as well and its director, Ian Hodder, for supporting our work and giving us the opportunity to contribute to the vast amount of research that this site has produced, and to the project coordinator Bilge Küçükdoğan for the technical support. Thanks to Jaime Pagán-Jiménez for the recommendations and advice, and to Zara Ali for proofreading this paper. A special thanks to CaSEs members that collected some of the material and to Teresa Garnatje, director of the Botanical Institute of Barcelona, and to Amanda Henry from Leiden University who kindly helped with expanding the UPF Laboratory of Environmental Archaeology reference collection. Christina Tsoraki would like to express her sincere thanks to: Prof. Annelou van Gijn, Director of the Laboratory for Material Culture Studies at Leiden University, for giving her permission to use the images of the experimental microwear reference collection; Weiya Li and other members of the Leiden laboratory for offering invaluable help with technical issues; and Prof. Alistair Pike, Head of the Department of Archaeology at the University of

Southampton, for facilitating access to lab facilities during summer 2020, when access to university buildings was restricted due to Covid-19.

## Author Contributions

**Conceptualization:** Carlos G. Santiago-Marrero, Christina Tsoraki, Marco Madella.

**Formal analysis:** Carlos G. Santiago-Marrero, Christina Tsoraki.

**Funding acquisition:** Christina Tsoraki, Carla Lancelotti, Marco Madella.

**Investigation:** Carlos G. Santiago-Marrero, Christina Tsoraki.

**Methodology:** Carlos G. Santiago-Marrero, Christina Tsoraki.

**Resources:** Christina Tsoraki, Carla Lancelotti, Marco Madella.

**Supervision:** Carla Lancelotti, Marco Madella.

**Validation:** Carlos G. Santiago-Marrero, Christina Tsoraki, Carla Lancelotti, Marco Madella.

**Visualization:** Carlos G. Santiago-Marrero, Christina Tsoraki, Carla Lancelotti, Marco Madella.

**Writing – original draft:** Carlos G. Santiago-Marrero, Christina Tsoraki, Carla Lancelotti, Marco Madella.

**Writing – review & editing:** Carlos G. Santiago-Marrero, Christina Tsoraki, Carla Lancelotti, Marco Madella.

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
