## [Decision Letter · Decision Letter 0]

17 Mar 2021

PONE-D-21-00753

A microbotanical and microwear perspective to plant processing activities and foodways at Neolithic Çatalhöyük.

PLOS ONE

Dear Dr Carlos Gabriel Santiago-Marrero,

Thank you for submitting your manuscript to PLOS ONE. After careful consideration, we feel that it has merit but does not fully meet PLOS ONE’s publication criteria as it currently stands. Therefore, we invite you to submit a revised version of the manuscript that addresses the points raised during the review process.

I have appended below the detailed comments of all three reviewers, who have recommended acceptance of your paper subject to revisions. I am requesting that you implement **all changes and edits as described by the reviewers** (and where this may not be possible, to provide a detailed rebuttal and/or justification for why this is the case). I wish to draw your attention particularly to the points raised by Reviewer 2 regarding the methodology and reporting of the results of use-wear analysis, and the issues identified by Reviewer 2 vis a vis the representativeness of the analysed sample, which (as a bare minimum) should be acknowledged and discussed explicitly in the text.

**Please submit your revised manuscript by 31st of March 2021**. If you will need more time than this to complete your revisions, please reply to this message or contact the journal office at plosone@plos.org. Please include the following items when submitting your revised manuscript:

We look forward to receiving your revised manuscript.

Kind regards,

Eleni Asouti, Ph.D.

Academic Editor

PLOS ONE

Journal Requirements:

3. Please include your tables as part of your main manuscript and remove the individual files. Please note that supplementary tables (should remain/ be uploaded) as separate "supporting information" files

Reviewers' comments:

Reviewer's Responses to Questions

**Comments to the Author**

1. Is the manuscript technically sound, and do the data support the conclusions?

Reviewer #1: Yes

Reviewer #2: Partly

Reviewer #3: Yes

2. Has the statistical analysis been performed appropriately and rigorously? 

Reviewer #1: Yes

Reviewer #2: N/A

Reviewer #3: N/A

3. Have the authors made all data underlying the findings in their manuscript fully available?

Reviewer #1: Yes

Reviewer #2: Yes

Reviewer #3: Yes

4. Is the manuscript presented in an intelligible fashion and written in standard English?

Reviewer #1: Yes

Reviewer #2: Yes

Reviewer #3: Yes

5. Review Comments to the Author

**Reviewer #1:** This paper is rigorous, thorough and interesting methodologically. It demonstrates how the analysis of micro-botanical proxies provides additional as well as complementary data to the analysis of the macro-botanical assemblage. The results presented here broaden our knowledge of the plants exploited at Catalhoyuk, particularly through the starch analysis. It would have been interesting if you had expanded on ethnographic examples of how the rhizomes and bulbs of Iridaceae and Liliaceae are usually processed and also of their use for medicinal purposes. You touch upon this but it is extremely interesting and so I think worthy of further discussion. For example, I know that in Chinese and Tibetan medicine, Iridaceae is used for its medicinal properties and is supposed to have an antibacterial effect (it is used against TB for example). Are there similar accounts of the use of Iridaceae in medicine in the Near East that could be relevant to this situation? I think including additional discussion on this topic would provide an additional dimension to the paper while specific detail on how these plants are processed would be useful in thinking of how the grinding stones were used.

On another note just a couple of edits:

Table 3 ‘smoot’ should presumably read ‘smooth’

Table 4 need to specify what R represents

**Reviewer #2: **This concise and very interesting article tackles the question of the function of the ground stone implements uncovered at the Neolithic site of Çatalhöyük. Although the collection has been previously presented in various papers, the question of tool function was mainly indirectly addressed so far, through morpho-typological analysis, using ethnographic analogies and/or spatial distribution. Thus, previous interpretations often posited grinding and pounding implements as a good proxy for plant processing and/or placed an emphasis on the multi-functional character of the tools. I believe that one of the great interests of the article is to go beyond such generalizations and to provide a direct, multi-approach investigation of ground stone tool function. The article presents in detail phytoliths and starch types extracted from the surface of a sample of tools. This is later compared with the results of a use-wear analysis, which are more briefly discussed. In the conclusion, the authors emphasize that their results provide additional evidence for a great diversity of plants being exploited at the site. The contribution of wild plants is also underlined, among which geophyte species and panicoid grasses which were seldom documented so far. Although these conclusions are well supported by the data presented, it is nevertheless based the analysis of a very limited sample of tools. In general, I believe that the authors should provide more information on their research and clarify various points so the significance and limits of the results presented can be more clearly assessed by the readers. Suggestions are provided bellow:

- One of the main points not addressed here is **sample size**. Use-wear and residue analyses are very time-consuming, still the fact that the study focuses on only 8 out of more than 2000 artifacts (yet less if we count only on grinding and pounding implements) should be more clearly acknowledged, discussed and taken into account in the discussion;

- Along the same line, the **sampling strategy** should be presented in more detail. From the description of the tools, it does not seem that the best-preserved artifacts were selected which is puzzling, especially because the methods employed encompass use-wear analysis.

- While the procedure, reference collection, limits in interpretation are well presented for phytoliths and starch, less details are provided for **use-wear approach**. More information about the reference collection used, some photographs of the experimental tools would be appreciated.

- Concerning **the results of phytoliths and starch analysis**: Were sediment samples only examined for starch? Is this because tool contamination was not found to be an issue for phytoliths? It is intriguing and noteworthy that sediment samples were very poor in starch while in situ plant processing is suggested for 2 of the buildings sampled here, I believe this should be discussed. It is my understanding that damages on starch are mainly attributed here to post-depositional processes, but this should be more clearly stated. The question of gelatinization and the fact that it may occur during post-depositional burning is very interesting. Are the authors referring here to the same type of starch grain damages than the ones ascribed in a few papers to beer production, or are those rather described as ‘full gelatinization’ which they do not found in Çatalhöyük ground stone tool sample? It would be useful to make this point clearer. Phytoliths modifications are not discussed while this could also have contributed to explore plant processing, the authors should specify this and maybe explain why.

- Regarding **the results of the use-wear analysis**: Given the state of alteration mentioned for some of the tools, an assessment on use-wear preservation should be provided. In several research, criteria have been suggested to differentiate various types of plants or processing for instance greasy/versus non greasy plants, cereals/legumes, dehusking/grinding (see Adams, Dubreuil, Hamon for example), yet here the authors are identifying ‘plant processing’. Why does the analysis remain at such a general level and do not further assess the properties of the matter processed? It may be due to the compressed versions of the images, I have but it is difficult to see any micropolish in Fig10 e and f. In addition, given the variability observed in use-wear associated with the processing of various types of plant, the concept of ‘plant micropolish’ is problematic (as there are different types of micropolish associated with plant processing). Furthermore, stone on stone contact has been reported to occur during grinding (see for instance Dubreuil 2004) so its development can be concomitant to other type of wear. Fig 10 B (and in the text as well) mention the processing of ‘ woody’ material, does the authors mean abrasion or grinding? It seems that the use-wear would fit with those associated with a use as an abrader but if the authors mean grinding, could they provide more explanation about the type of woody matter and for what purpose? Finally, it seems that part of the tool life-history is described in the section presenting the artifacts, with no or minimal supporting information regarding how those conclusions were reached, while in fact this life-history reconstruction should be part of the use-wear analysis.

- Overall, the article is well illustrated.** One table **with synthetic presentation of the residues and use-wear results for each stone is missing, so both approaches could really be integrated. **Providing more illustration **of use-wear, of the experimental collection and surfaces of the tool at low magnifications would also strengthen the interpretation suggested. Finally, I would suggest including **a few figures** combining the various faces of the tool, use-wear and residues found on the same object.

**Reviewer #3: **This is a very clearly written and exciting paper that reports on the first integrated phytolith, starch and use-wear results from ground stone implements at Catalhoyuk. The results are explained with admirable clarity and with full reference to the relevant literature. I have only one, relatively minor, comment. Regarding the Panicoid grasses and C4 isotope discussion, the macrobotanical assemblage does contain some other C4 grasses (Aeluropus, Crypsis, Sporobolus) that were also observed in intact sheep/goat pellets, and so likely contribute to both the caprine C4 signal and the human C4 signal. This does not exclude that other, Panicoid grasses also contribute to the human C4 signal, but it should be made clear in the text that there are other C4 taxa that likely contributed in any case.

6. PLOS authors have the option to publish the peer review history of their article (what does this mean?). If published, this will include your full peer review and any attached files.

Reviewer #1: **Yes: **Emma Louise Jenkins

Reviewer #2: No

Reviewer #3: No

---

## [Author Response · Author response to Decision Letter 0]

5 May 2021

Reviewer #1: 

This paper is rigorous, thorough and interesting methodologically. It demonstrates how the analysis of micro-botanical proxies provides additional as well as complementary data to the analysis of the macro-botanical assemblage. The results presented here broaden our knowledge of the plants exploited at Catalhoyuk, particularly through the starch analysis. 

• It would have been interesting if you had expanded on ethnographic examples of how the rhizomes and bulbs of Iridaceae and Liliaceae are usually processed and also of their use for medicinal purposes. You touch upon this but it is extremely interesting and so I think worthy of further discussion. 

We agree that ethnographic examples are extremely interesting and deserve to be further discussed. However, we are also aware of the possible problems that could emerge by trying to make more direct parallels between modern-day uses of these plants and the possible practices performed at Çatalhöyük. Despite that it has been demonstrated that many of these plants are beneficial for many health aspects, the reasons behind the use and the value attributed to a particular medicinal plant are also part of a cultural construction. Being aware of this, we decided to be conservative in our interpretations, even more so when considering the 8,000 years gap between the material studied and the ethnographic examples. For this reason, we just limited our discussion to the possible seasonal use, since these are directly related to the plant's life circles, and from there, we use the ethnographic examples to demonstrate that these are indeed important resources and could have been in the past as well.

• For example, I know that in Chinese and Tibetan medicine, Iridaceae is used for its medicinal properties and is supposed to have an antibacterial effect (it is used against TB for example). Are there similar accounts of the use of Iridaceae in medicine in the Near East that could be relevant to this situation? 

Indeed, there are similar accounts for the use of Iridaceae species for medicinal purposes in the Near East, and particularly in Turkey. The information regarding which Iridaceae species has been used in medicine can be found in Table 5, section 4.1. More information about the specific uses is available on the cited literature in the same table.

• I think including additional discussion on this topic would provide an additional dimension to the paper while specific detail on how these plants are processed would be useful in thinking of how the grinding stones were used.

Despite the rich available literature (at least the one written in English) about the uses of Iridaceae species, many of the consulted papers only report how these plants were consumed, and the parts of interest. From these accounts, it can be deduced that its consumption involves a process like pulverizing and mashing but these were not explicitly exposed. However, soon after submitting the manuscript, we found a paper that does include explicit processing like crushing and powdering of Liliaceae bulbs that can be performed with grinding and crushing tools [1]. We have made a short mention in the test of these plants, their medicinal uses and how these are processed.

Section 4.1, lines 575- 579.

It is worth noting that some Liliaceae and Iridaceae species contain toxins and need to be processed for detoxification before consumption, or they are used in small quantities for medicinal purposes. This is the case for species of the genus such as Fritillaria, Lilium and Tulipa, whose USOs can be crushed and powdered and used in folk medicine in parts of current Turkey for treating wounds and other ailments [190,193].

Regarding how these could have been processed in the past, many of the underground storage organs mentioned in the paper can be consumed or processed fresh. But, considering the hardness of some of these, a pounding, rather than grinding process could be assumed. However, this is not always the case, underground storage organs can be dried first, and then ground, such as the case of orchid bulbs used for making Salep, a starchy flour commonly consumed in Turkey in beverages and desserts. Other practices include boiling, roasting, or even fermenting that make the tubers softer thus removing the need for pounding. These practices would affect the way such plants were processed and the kind of reduction technique and tools used. 

We ignore whether these practices were also performed at the site, and in the case of underground storage organs, and despite the obvious assumptions that can be made based on the tools' typology, the starch data can only tell us that these plants were subject to a reduction technique which involved the tool where the starch was found. Besides, there is also an etymological consideration involved in the way we interpret the tools uses, and the reduction technique that was used at the time of processing the USO. This is related to the intrinsic definition of a grinding stone itself, that links an object type with a very specific practice. We are aware that this is not always the case, these are dynamic objects, involved in multiple functions and reduction techniques that expand the plant-related activities as it has been demonstrated in its section. 

In sum, we recognize the great interpretative potential that can emerge by going into detail about the plant's characteristics that influence the ways it could be treated, and the reduction technique used. But we are also aware that the information provided by the tools is a palimpsest of multiple plants, uses, and activities, and based on the arguments presented above we prefer to be careful and to not overextend the interpretative potential of our finds. 

• On another note just a couple of edits:

• Table 3 ‘smoot’ should presumably read ‘smooth’

• Table 4 need to specify what R represents

These mistakes and editions have been addressed. 

Reviewer #2:

 This concise and very interesting article tackles the question of the function of the ground stone implements uncovered at the Neolithic site of Çatalhöyük. Although the collection has been previously presented in various papers, the question of tool function was mainly indirectly addressed so far, through morpho-typological analysis, using ethnographic analogies and/or spatial distribution. Thus, previous interpretations often posited grinding and pounding implements as a good proxy for plant processing and/or placed an emphasis on the multi-functional character of the tools. I believe that one of the great interests of the article is to go beyond such generalizations and to provide a direct, multi-approach investigation of ground stone tool function. The article presents in detail phytoliths and starch types extracted from the surface of a sample of tools. This is later compared with the results of a use-wear analysis, which are more briefly discussed. In the conclusion, the authors emphasize that their results provide additional evidence for a great diversity of plants being exploited at the site. The contribution of wild plants is also underlined, among which geophyte species and panicoid grasses which were seldom documented so far. Although these conclusions are well supported by the data presented, it is nevertheless based the analysis of a very limited sample of tools. In general, I believe that the authors should provide more information on their research and clarify various points so the significance and limits of the results presented can be more clearly assessed by the readers. Suggestions are provided bellow:

• One of the main points not addressed here is sample size. Use-wear and residue analyses are very time-consuming, still the fact that the study focuses on only 8 out of more than 2000 artifacts (yet less if we count only on grinding and pounding implements) should be more clearly acknowledged, discussed and taken into account in the discussion;

We recognize the concern about the sample size. However, the artifacts were selected because when we started this study, there were among the most complete artifacts since complete tools are not that common on the site (see on section buildings and material). In addition, most of the tools were extensively handled, minimizing the sample size due to the risk of some artifacts have being exposed to modern starch contamination. We also realized that these artifacts were found on very important and well-studied buildings and contexts providing an opportunity to integrate our results into further and multidisciplinary discussions about the activities performed on buildings 80, 52 and 44. Nonetheless, despite being a small sample, we believe that by implementing a multi-proxy approach, we produced a more robust and complex set of data than the one we would have obtained by analyzing more artifacts with only one proxy.

To conclude, we believe that the small sample size does not hamper the informative potential of these artifacts as has been highlighted in the paper. The main point of the paper was not to generalize on how grinding tools were used at the site but to demonstrate the research potential of combining two microbotanical proxies with microwear analysis. We believe that this was successfully achieved based on the diversity of plants identified, including wild plants processed on-site that had not been previously identified in the macrobotanical record. In this sense, this is not an attempt to deduce the scale of processing and reliance on these plant resources nor variations between periods or different buildings, which would require a more robust sample. But we recognize the need of acknowledging the sample size as suggested and has been addressed in the paper.

Section 5.0, lines 770 – 772.

Despite that the tools examined represent only a small portion of the whole grinding tool assemblage at the site, our work made evident the wide spectrum of plant resources, both domesticated and wild, that the people of Çatalhöyük were using.

Further discussion below.

• Along the same line, the sampling strategy should be presented in more detail. From the description of the tools, it does not seem that the best-preserved artifacts were selected which is puzzling, especially because the methods employed encompass use-wear analysis.

Some of the reason that influenced the sampling criteria has been presented above. But we have made changes addressed this on the paper.

Section 2.4, lines 277 - 287

Following the technological analysis of the ground stone assemblage a number of objects were selected for microwear analysis, including the set of tools that were further sampled for microbotanical analysis. The combination of microwear and microbotanical analyses discussed in this paper allowed not only for a better understanding of the plant exploitation strategies but also for a more nuanced appreciation of tool function at the site [89,90]. The selection of material was guided by typological, technological and contextual criteria, as well as the overall condition of the material. While tools that survive in generally good condition without visible alterations were favoured, in some cases tools that show low or moderate heat alterations but derive from significant contexts (e.g., fixed grinding installation 21767.x2 from B.80) and/or survive intact were also included in the sampled material. In cases where the tools exhibited more intense burning, which affected the use-faces of the implements, these were not subjected to microwear analysis. 

• While the procedure, reference collection, limits in interpretation are well presented for phytoliths and starch, less details are provided for use-wear approach. More information about the reference collection used, some photographs of the experimental tools would be appreciated.

We created a separate section on Microwear analysis in Materials and Methods (section 2.4, lines 277-334), where we included more information about microwear analysis, the experimental reference collection, and created a new figure with experimental wear traces.

• Concerning the results of phytoliths and starch analysis: Were sediment samples only examined for starch? 

Each sediment sample was subjected to both, starch and phytolith analyses as has been presented in the section 2.2 Sampling and laboratory procedures for microbotanical analysis.

Section 2.2, lines 250 – 253.

The extraction procedure followed a combined protocol for phytoliths and starch extraction adapted from Pagan-Jimenez [59] and García-Granero et al. [112]. In this method, starch grains are first extracted from the sediments to avoid their exposure to any stronger chemical used for the phytolith extraction [113–115].

We also modified the Taphonomy and contamination section accordingly.

Section 3.1.2, lines 469 – 470.

Control sediment samples were rich in phytoliths and devoid of starch except for Building 44 where the sediment produced 9 heavily damaged starch grains.

• Is this because tool contamination was not found to be an issue for phytoliths? 

Despite this being a problem that applies to both proxies, there is always more concern about the starch so much so that a few researchers have doubts over the preservation of starch in archaeological sediments/artefacts. Therefore, when working with ancient starch there is always concerns whether the starch recovered is ancient or a result of modern contamination. This topic has been widely discussed, but the assumption is that ancient starch survived trapped in the tool's porous surface. This, assumption is supported by the fact that the control sediments do not present starch granules or only a few. 

Besides the check provided by control sample, we proposed a novel solution by correlating the most predominant starch damage with the tool's depositional or post-depositional context. To be more precise, those that were exposed to fire vs those that were not.

• It is intriguing and noteworthy that sediment samples were very poor in starch while in situ plant processing is suggested for 2 of the buildings sampled here, I believe this should be discussed. 

In-situ plant processing was only suggested for the quern embedded in the floors of building 80 because this was a fixed installation. We discussed this artifact and the interpreted in-situ plant processing in section 4.3.1. Artifacts from building 44 were deposited in a cluster and those from building 52 were in a storage room, thus we cannot nor are suggesting that these were used in-situ and their mobile nature has been discussed on the paper as well in Section 4.3.2. 

We believe the confusion could be caused by sentence on the original manuscript;

This in situ assemblage related to plant storage suggests a diversified diet [21], and the spectrum of plant use in this building was probably even wider than what the macrobotanical evidence suggests [80].

We have modified this sentence;

Section 2.1, line 185 - 188

This assemblage related to plant storage suggests a diversified diet [21], and the spectrum of plant use in this building was probably even wider than what the macrobotanical evidence suggests [80].

This was also addressed and clarified in Section 4.3.2, lines 666 – 669.

The very low presence of starch grains on the tool (n=11) and the complete absence of starch from the surrounding sediments (see [246]), together with the phytolith evidence, strongly support that this implement was used in situ for dehusking or beating spikelets for separating seeds from the chaff in cereals (see [10,247,248]).

• It is my understanding that damages on starch are mainly attributed here to post-depositional processes, but this should be more clearly stated. 

Concerning the damage on starch as being caused by post-depositional process (fire), it is mentioned in the paper, and its interpretation is only limited to strengthen its origin as ancient starch. 

Section 3.1.2, line 489 – 490.

These observations are useful as they confirm that the recovered starches are of secure provenance and not the product of modern contamination.

Further discussion below.

• The question of gelatinization and the fact that it may occur during post-depositional burning is very interesting. Are the authors referring here to the same type of starch grain damages than the ones ascribed in a few papers to beer production, or are those rather described as ‘full gelatinization’ which they do not found in Çatalhöyük ground stone tool sample? It would be useful to make this point clearer. 

Starch damage can indeed be very useful to identify certain practices such as fermenting, boiling, etc. It's possible that some of the observed damage was caused by the type of processing to which the plants were subject, such as the case for broken or cracked starch grains, traces that occur while grinding and pounding. I believe here the confusion could be cause by what is causing the starch gelatinization either partially or full. Heat treatment is the principal anthropic cause of gelatinization on starch, in our case, this could be caused by processing practices such as roasting or parching the grains before grinding. However, enzymatic activity can also cause starch gelatinization, produced when the grains are left to sprout (intentionally or unintentionally) before being processed or by microbial activity (fermenting). However, making such interpretation is very complex, not all starches are equally resistant to heat, enzymes, or mechanical damage and we do not have enough information to support any of the mentioned treatments, especially since some artifacts were exposed to fire. In addition, apart from differentiating from the heat environment (wet or dry), starch gelatinization can not be exclusively linked to a specific treatment [2–4]. Instead, its interpretation relies vastly on the archaeological context [4,5].

Concerning our interpretation of the exposure to fire as the cause of starch gelatinization, we consider that this exposure restrict any attempt to link any observed starch damage to a particular processing activity. Therefore, we can only hypothesize that starch grains that were already trapped in the tools started to partially gelatinize when the artifacts were exposed to heat once the buildings 80 and 52 burned down. 

Following the reviewer suggestion, we have made this point clearer. 

Section 3.2, lines 486 – 489.

Therefore, gelatinization-like damage in our samples in all probability is not related to intentional food processing activities or cooking, but to post-depositional processes such as the artefact’s exposure to heat when both buildings caught fire. 

• Phytoliths modifications are not discussed while this could also have contributed to explore plant processing, the authors should specify this and maybe explain why.

This is a very important observation. Despite that it is not as informative or evident as in the case of starch, certain mechanical processes like threshing or dehusking could cause straight cuts or breaks on silicified conjoined cells. These could be used as a proxy for post-harvesting processes. We have made changes to the text and included this.

Section 3.2, lines 526 – 527.

In some cases, silica skeletons presented straight transversal cuts (Fig 10. B-C), probably caused by post-harvesting processes such as trampling and dehusking [175,176].

• Regarding the results of the use-wear analysis: Given the state of alteration mentioned for some of the tools, an assessment on use-wear preservation should be provided. 

Clarification about heat alterations was provided in Section 2.1.and in section 2.4. As noted in Table 1 the two tools with moderately/heavy degree of burning were not subjected to microwear analysis. 

• In several research, criteria have been suggested to differentiate various types of plants or processing for instance greasy/versus non greasy plants, cereals/legumes, dehusking/grinding (see Adams, Dubreuil, Hamon for example), yet here the authors are identifying ‘plant processing’. Why does the analysis remain at such a general level and do not further assess the properties of the matter processed? It may be due to the compressed versions of the images, I have but it is difficult to see any micropolish in Fig10 e and f. In addition, given the variability observed in use-wear associated with the processing of various types of plant, the concept of ‘plant micropolish’ is problematic (as there are different types of micropolish associated with plant processing). Furthermore, stone on stone contact has been reported to occur during grinding (see for instance Dubreuil 2004) so its development can be concomitant to other type of wear.

We included a paragraph in Section 2.4, lines 309 - 330 that discusses the difficulty in differentiating types of plants in archaeological materials that display mixed wear signatures and are associated with the processing of different contact materials and have included in section 4.0 Discussion more information about the type of plant processing when this level of interpretation was possible. In the case of tools with evidence for use against multiple materials and mixed wear signatures it is not possible to interpret the type of plant contact material in more detail. We agree about the stone on stone contact which we also have identified during the microwear analysis of some of the grinding tools (see Table 1) and we clarified this point further in section 4.3.2, lines 707 – 709.

• Fig 10 B (and in the text as well) mention the processing of ‘ woody’ material, does the authors mean abrasion or grinding? It seems that the use-wear would fit with those associated with a use as an abrader but if the authors mean grinding, could they provide more explanation about the type of woody matter and for what purpose? 

We addressed this point in Section 4.3.2 and lines 736-744. 

In the case of the woody plant materials, based on the characteristics of the use-face of the tools (intense levelling and smoothing that extends across the whole surface of the tool), their presence on the tool surfaces is more consistent with grinding activities. The presence of wood and wood-like materials has also been noted in other studies of grinding tools [35,137]. Hamon and colleagues [137] have suggested that the grinding of wood and bark materials may be associated with the acquisition of powdered materials with colouring or medicinal properties, or potentially fruit processing. Continuing microwear analysis of the Çatalhöyük grinding tools may shed more light on the type of processing associated with this type of contact material.

• Finally, it seems that part of the tool life-history is described in the section presenting the artifacts, with no or minimal supporting information regarding how those conclusions were reached, while in fact this life-history reconstruction should be part of the use-wear analysis.

The detailed technological, spatial and contextual analysis of the Çatalhöyük ground stone assemblage has been presented in other publications and we included in the text in section 2.1. the following passage to reflect this: 

Line 134 - 137: Detailed technological, typo-morphological and contextual study of the Çatalhöyük ground stone assemblage, the results of which are presented in detail elsewhere [88,90], provides important insight in the life histories of these artefacts and the social practices that surround their use and discard by this Neolithic community.

In addition, we have provided some additional information about the tools in the text in section 2.1. which should be read in tandem with the information given in Table 1.

• Overall, the article is well illustrated. One table with synthetic presentation of the residues and use-wear results for each stone is missing, so both approaches could really be integrated. Providing more illustration of use-wear, of the experimental collection and surfaces of the tool at low magnifications would also strengthen the interpretation suggested. Finally, I would suggest including a few figures combining the various faces of the tool, use-wear and residues found on the same object.

We appreciate the suggestion and have included a collage picture showing the artifact along with use-wear traces and micro-botanical remains on Section 4.3.2, line 714.

Reviewer #3: 

This is a very clearly written and exciting paper that reports on the first integrated phytolith, starch and use-wear results from ground stone implements at Catalhoyuk. The results are explained with admirable clarity and with full reference to the relevant literature. 

• I have only one, relatively minor, comment. Regarding the Panicoid grasses and C4 isotope discussion, the macrobotanical assemblage does contain some other C4 grasses (Aeluropus, Crypsis, Sporobolus) that were also observed in intact sheep/goat pellets, and so likely contribute to both the caprine C4 signal and the human C4 signal. This does not exclude that other, Panicoid grasses also contribute to the human C4 signal, but it should be made clear in the text that there are other C4 taxa that likely contributed in any case.

We agree on this observation and changes has been made on the text on Section 4.2, lines 635 - 646).

Bibliography

1. Bozyel ME, Merdamert-Bozyel E, Benek A, Canlı K, Altuner EM. Ethnomedicinal Uses of Colchicaceae and Liliaceae Taxa in Turkey. In: Christov I, Krystev V, Efe R, editors. Advances in Scientific Research: Engineering and Architecture. Sofia: St. Kliment Ohridski University Press; 2020. pp. 649–660. 

2. Pagán-Jiménez J. Evaluando algunos mecanismos de conservación/degradación en almidones modernos por medio de ensayos y experimentos controlados que replican ciertas formas antiguas de procesamiento y cocción de órganos almidonosos. Elaboración de dos tipos de chicha de maíz: chicha fermentada con saliva y otra con levadura. Quito: Instituto Nacional de Patrimonio Cultural; 2015. doi:10.13140/RG.2.1.5089.7440

3. Pagán‐Jiménez JR, Guachamín-Tello AM, Romero-Bastidas ME, Vásquez-Ponce P. Cocción experimental de tortillas de casabe (Manihot esculenta Crantz) y de camote (Ipomoea batatas [L.] Lam.) en planchas de barro: evaluando sus efectos en la morfometría de los almidones desde una perspectiva paleoetnobotánica. Americae. 2017;2: 27–44. 

4. Chantran A, Cagnato C. Boiled, fried, or roasted? Determining culinary practices in Medieval France through multidisciplinary experimental approaches. J Archaeol Sci Rep. 2021;35: 102715. doi:10.1016/j.jasrep.2020.102715

5. García-Granero JJ. Starch taphonomy, equifinality and the importance of context: Some notes on the identification of food processing through starch grain analysis. J Archaeol Sci. 2020;124: 105267. doi:10.1016/j.jas.2020.105267

---

## [Editor Report · Decision Letter 1]

14 May 2021

A microbotanical and microwear perspective to plant processing activities and foodways at Neolithic Çatalhöyük.

PONE-D-21-00753R1

Dear Carlos Gabriel Santiago-Marrero,

We’re pleased to inform you that your manuscript has been judged scientifically suitable for publication and will be formally accepted for publication once it meets all outstanding technical requirements.

Kind regards,

Eleni Asouti, Ph.D.

Academic Editor

PLOS ONE

Additional Editor Comments (optional):

Please ensure that your tables are adequately formatted for inclusion in the manuscript as it will appear on the journal website and in pdf form. I still have some concerns regarding the size and formatting of Table 1, including its individual cells (parts of the text are missing and/or are illegible in the .pdf of Table 1 you submitted as a separate attachment). Please read very carefully all the instructions included in the author guidelines and liaise closely with the PLOS ONE editorial team. It will not be possible to fix any formatting issues resulting in parts of Tables being cropped or illegible after publication. Please also note that it is your responsibility to proof read thoroughly all text, tables and captions to correct any residual typos, when you receive your proofs from the PLOS ONE journal office.

---

## [Editor Report · Acceptance letter]

2 Jun 2021

PONE-D-21-00753R1 

A microbotanical and microwear perspective to plant processing activities and foodways at Neolithic Çatalhöyük. 

Dear Dr. Santiago-Marrero:

I'm pleased to inform you that your manuscript has been deemed suitable for publication in PLOS ONE. Congratulations! Your manuscript is now with our production department. 

Kind regards, 

on behalf of

Dr. Eleni Asouti 

Academic Editor

PLOS ONE